# FDTDNet: Lensless Object Segmentation via Feature Demultiplexing and Task Decoupling

## Abstract

Camera-based vision systems pose privacy risks, whereas lensless cameras present a viable alternative by omitting visual semantics from their measurements due to the absence of lenses. However, these captured lensless measurements pose challenges for existing computer vision tasks such as object segmentation that usually require visual input. To address this problem, we propose a lensless object segmentation network via feature demultiplexing and task decoupling (FDTDNet) to perform object segmentation for lensless measurements. Specifically, we propose an optical-aware feature demultiplexing mechanism to get meaningful features from lensless measurements without visual reconstruction and design a multi-task learning framework decoupling the lensless object segmentation task into two subtasks, i.e., the reason for contour distribution maps (CDM) and body distribution maps (BDM), respectively. Extensive experiments demonstrate that our FDTDNet achieves highly accurate segmentation effect, which sheds light on privacy-preserving high-level vision with compact lensless cameras.

## 1 Introduction

Lensless cameras (Tan et al. (2019); Pan et al. (2021b;a); Salman et al. (2022)) utilize simple, planar optics to convert light into complex patterns, rendering the images unintelligible without knowledge of the mask configurations. Their enhanced privacy features make them promising for privacy-focused applications (Pan et al. (2021b); Yin et al. (2022)). For object segmentation in Fig. 1(a), traditional systems use converging lenses to capture clear images before applying segmentation algorithms, making them vulnerable to network attacks. Lensless cameras produce ambiguous measurements that help safeguard sensitive information (Pan et al. (2021b); Yin et al. (2022)). The typical lensless method (You et al. (2022)) involves restoring the image using the mask, followed by conventional segmentation, as illustrated in Fig. 1(b). However, this method has drawbacks: it prevents data hijacking but is still susceptible to software attacks on reconstructed images. Additionally, segmentation accuracy suffers from blurry reconstructions and suboptimal mask designs Yin et al. (2022), while the reconstruction adds computational overhead, making it less suitable for edge computing.

To enhance segmentation accuracy while ensuring privacy, we propose a one-step method for lensless object segmentation. Unlike the classical two-step process, our method directly segments objects from lensless measurements without intermediate reconstructions, as shown in Fig. 1(c). However, extracting sufficient semantic features in the absence of visual input presents a severe challenge.

To overcome this limitation, we propose an optical-aware feature demultiplexing (OFD) mechanism aimed at refining the features obtained from lensless measurements. This concept is underpinned by the observation that lensless measurements exhibit a direct linear correlation with visual images through the measurement matrix. Similarly, the semantic features corresponding to these measurements can be represented through a linear relationship based on the semantic attributes of the visual images. Considering this, we define $Y$ as the lensless measurement, $A$ as the measurement matrix, and $X$ as the original image. Correspondingly, $Y_{\theta_i}$, $A_\theta$, and $X_{\theta_i}$ denote the associated semantic features. Inspired by (Dong et al. (2021)), the above relationship can be succinctly articulated by the following linear equation,

$$Y = A \circ X \implies Y_{\theta_i} = A_\theta \circ X_{\theta_i}. \tag{1}$$

Building upon this correlation, we propose a feature demultiplexing and task decoupling network (FDTDNet) for lensless object segmentation. Our approach reconstructs desired features $X_{\theta_i}$ from

Figure 1: Comparison of object segmentation methods: traditional lens-based (top), two-step lens-less (middle), and our optimized one-step lensless methods. Ours improves privacy protection against data interception and software vulnerabilities while maintaining robust segmentation effect.

semantic features $Y_{\theta_i}$ while preserving privacy, utilizing the OFD mechanism. By integrating OFD with a Pyramid Vision Transformer (PVT), we enhance long-range feature extraction to tackle segmentation challenges. We further decouple segmentation labels into a contour distribution map (CDM) and a body distribution map (BDM) to mitigate imbalanced pixel distribution issues. To facilitate effective aggregation of CDM and BDM, we introduce a mutual learning strategy using the contour-body interaction (CBI) module. Our main contributions are as follows:

• To our best knowledge, we investigate direct object segmentation from lensless measurements and propose a high-accuracy lensless object segmentation method, which verifies the potential of applying lensless imaging directly to various high-level tasks.

• We model the linear equation between the semantic features bound to lensless measurements and those corresponding to visual inputs. By the proposed OFD, we obtain the expected semantic features to enhance prediction performance.

• We decouple the segmentation task into CDM and BDM inference by contour-/body-distribution learning branches. And a contour-body interaction (CBI) module is proposed for reasoning segmentation results from correlations between CDM and BDM.

• Extensive experiments on two datasets (*i.e.*, directly captured (DIRC) dataset and display captured (DISC) dataset) indicate that our FDTDNet outperforms state-of-the-art methods by a large margin.

## 2 RELATED WORKS

### 2.1 LENSLESS IMAGING

Lensless imaging (M. Salman et al. (2017); Nick et al. (2018); Pan et al. (2022); Jiachen et al. (2020)) provides an effective way to handdle size constraints in areas like smartphone photography and micro-robotics, relying on masks with amplitude and phase encoding as key components. Different mask architectures have driven the creation of prototypes such as the Fresnel Zone Aperture (FZA) camera (Jiachen et al. (2020); Wu et al. (2021)), FlatCam (M. Salman et al. (2017)), PhlatCam (Nick et al. (2018)), and DiffuserCam (Vivek et al. (2020)). These prototypes have proven valuable in areas such as hyperspectral imaging (Monakhova et al. (2020)), fluorescence microscopy (Alok et al. (2017)), light field encoding Tajima et al. (2017); Cai et al. (2020), and depth sensing (Nick et al. (2018); Tian & Yang (2022)). Recently, researchers have expanded lensless imaging to high-level semantic tasks, successfully achieving recognition (Pan et al. (2021a); Tan et al. (2019); Zhang et al. (2022); Aschenbrenner et al. (2024)), face verification (Tan et al. (2019); Cai et al. (2024)), and object segmentation (Yin et al. (2022; 2024)), showing its potential for high-level inference tasks.

## 2.2 RECONSTRUCTION-FREE SEMANTIC INFERENCE

Reconstruction-free semantic inference has attracted significant attention, finding applications in fields such as biomedicine, agriculture, and non-visual recognition (Lei et al. (2019); Isogawa et al. (2020); Qiu et al. (2024)). This kind of method offers key benefits in terms of privacy-preserving and reduced computational costs, especially in image recognition (Dave et al. (2022); Hinojosa et al. (2022)). In single-pixel cameras, it enhances computational efficiency (Ji et al. (2022); Liu et al. (2023)), and in lensless cameras, it enables tasks like classification directly from raw measurements (Cai et al. (2024); Perez et al. (2024); Yang et al. (2024)). Recent research has focused on pixel-level reasoning tasks like image segmentation (Yang et al. (2022)). In (You et al. (2022)), human eye segmentation was studied using a reconstruction-before-segmentation approach, but the high computational cost limited practical use. The works (Yin et al. (2022; 2024)) introduced an end-to-end network for segmenting objects from lensless imaging data, but its performance was constrained by the need for original scene supervision. Thus, achieving high-precision segmentation in lensless imaging remains challenges.

# 3 METHODOLOGY

## 3.1 MOTIVATION AND OVERVIEW

Among various lensless camera prototypes, FlatCam (M. Salman et al. (2017)) stands out for its wide range of applications due to its high luminous flux, lightweight setups, and cost-effectiveness. We investigate object segmentation based on the FlatCam imaging model, although our method is also easily adaptable to other lensless camera models. FlatCam utilizes a separable mask pattern, *i.e.*, the 2-D mask pattern, which can be represented by the outer product of two 1-D patterns. The imaging model is formulated as

$$Y = A_{\mathrm{L}} X A_{\mathrm{R}}^{\top} + \xi, \tag{2}$$

where $A_{\mathrm{L}} \in \mathcal{R}^{V \times M}$ and $A_{\mathrm{R}}^{\top} \in \mathcal{R}^{N \times W}$ denote matrices that correspond 1D convolutions along the rows and columns, respectively. The $\xi$ repsents the additive noise.

Lensless cameras produce multiplexed measurements $Y$ devoid of visual information, complicating object segmentation. To tackle this, we propose FDTDNet, a lensless object segmentation method leveraging feature demultiplexing and task decoupling. We first develop an optical-aware feature demultiplexing (OFD) mechanism integrated with a Pyramid Vision Transformer (PVT) for semantic feature decoupling. Then, the segmentation task is divided into contour distribution map (CDM) and body distribution map (BDM) subtasks, enhancing edge feature learning and reducing interference. We implement a contour-distribution learning branch with a dual-path attention (DPA) and a body-distribution learning branch with contextual exploration (CE) and hierarchical information fusion (HIF) for CDM and BDM predictions, respectively. A cross-branch learning strategy via the contour-body interaction (CBI) module further improves segmentation by exploiting the correlation between CDM and BDM.

## 3.2 OPTICAL-AWARE FEATURE DEMULTIPLEXING (OFD) MECHANISM

Unlike previous works that extract features from natural images, our encoder derive cues from lensless measurements, making conventional encoders ineffective. Thus, we propose a feature demultiplexing mechanism by integrating the OFD at the end of the PVT encoder to mine high-level information. First, we utilize the PVT for feature extraction in lensless object segmentation, generating outputs across four stages:

$$Y_{\theta_1}, Y_{\theta_2}, Y_{\theta_3}, Y_{\theta_4} = \mathrm{PVT}\,(Y)\,. \tag{3}$$

Based on Eq. (1) and the lensless imaging model in Eq. (2), the above semantic features $Y_{\theta_i}$ ($i = 1, 2, 3, 4$) is modeled as:

$$Y_{\theta_i} = A_{\mathrm{L},\theta_i} X_{\theta_i} A_{\mathrm{R},\theta_i}^{\top} + \xi, \tag{4}$$

where $X_{\theta_i}$, $A_{\mathrm{L},\theta_i}$, and $A_{\mathrm{R},\theta_i}$ denote the $X$, $A_{\mathrm{L}}$, and $A_{\mathrm{R}}$ in the feature space. Therefore, the task of reasoning about $X_{\theta_i}$ from $Y_{\theta_i}$ can be modeled as an inverse problem. To obtain $X_{\theta_i}$ for boosting the lensless object segmentation task, inspired by (Salman et al. (2022)), our OFD-based extractor is designed as the Tikhonov regularization problem as:

$$\arg\min_{X_{\theta_i}} \left\| Y_{\theta_i} - A_{\mathrm{L},\theta_i} X_{\theta_i} A_{\mathrm{R},\theta_i}^{\top} \right\|_2^2 + K_{\theta_i} \left\| X_{\theta_i} \right\|_2^2, \tag{5}$$

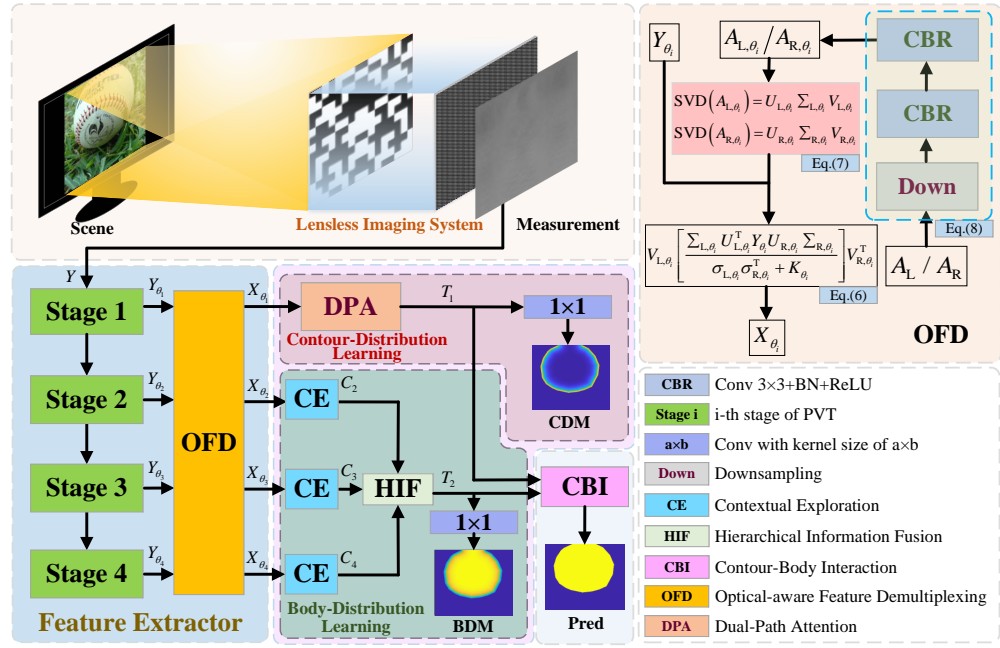

Figure 2: The proposed FDTDNet framework includes: (1) a PVT and OFD-based extractor for reconstructing semantics, (2) a contour-distribution learning branch with the DPA, and a body-distribution learning branch with the CE and HIF for inferring CDM and BDM, respectively, and (3) a CBI-based mutual learning strategy to derive segmentation results from CDM and BDM.

where $K_\theta$ is the learnable regularization parameter. The Eq. (5) can be sovlved by Wiener deconvolution (Haywood & Younes (2023)) as:

$$\hat{X}_{\theta_i} = \text{OFD}(Y_{\theta_i}; A_{\text{L},\theta_i}, A_{\text{R},\theta_i})$$
$$= V_{\text{L},\theta_i}[(\Sigma_{\text{L},\theta_i} U_{\text{L},\theta_i}^\top Y_{\theta_i} U_{\text{R},\theta_i} \Sigma_{\text{R},\theta_i})./(\sigma_{\text{L},\theta_i} \sigma_{\text{R},\theta_i}^\top + K_{\theta_i})]V_{\text{R},\theta_i}^\top, \tag{6}$$

satisfies with

$$A_{\text{L},\theta_i} \stackrel{\text{SVD}}{=} U_{\text{L},\theta_i}\Sigma_{\text{L},\theta_i}V_{\text{L},\theta_i}^\top, \quad A_{\text{R},\theta_i} \stackrel{\text{SVD}}{=} U_{\text{R},\theta_i}\Sigma_{\text{R},\theta_i}V_{\text{R},\theta_i}^\top, \tag{7}$$

where SVD is the singular value decomposition (SVD). The $A_{\text{L},\theta_i}$ and $A_{\text{R},\theta_i}$ are updated by

$$A_{\text{L},\theta_i} = f_{A_\text{L}}(A_\text{L}), A_{\text{R},\theta_i} = f_{A_\text{R}}(A_\text{R}), \tag{8}$$

where $f_{A_\text{L}}(\cdot)$ and $f_{A_\text{R}}(\cdot)$ represent the $3 \times 3$ convolution layer + batch normalization (BN) + ReLU + down-sampling operator. We denote each side output as $X_{\theta_i}$ for $\hat{X}_{\theta_i}$. The detailed derivation of the OFD is illustrated in Appendix A.1. Importantly, the OFD mechanism facilitates back-end tasks without visual reconstruction, mitigating sensitive privacy leakage.

### 3.3 TASK DECOUPLING

We assume the contour distribution follows a Gaussian distribution with zero mean and a standard deviation of $\sigma$. The ideal contour distribution and body distribution are defined as follows:

$$\mathcal{P}_{\text{cdm}}(\mathbf{p}) = \frac{1}{\sqrt{2\pi}\sigma}e^{-\frac{(\text{DT}(\mathbf{p}))^2}{2\sigma^2}}, \quad \mathcal{P}_{\text{bdm}}(\mathbf{p}) = 1 - \frac{1}{\sqrt{2\pi}\sigma}e^{-\frac{(\text{DT}(\mathbf{p}))^2}{2\sigma^2}}, \tag{9}$$

where $\sigma = \frac{1}{\sqrt{2\pi}}$ to keep $\mathcal{P}_{\text{cdm}}$ and $\mathcal{P}_{\text{bdm}}$ in the range [0,1]. $\text{DT}(\mathbf{p})$ represents the shortest Euclidean distance from pixel $\mathbf{p}$ to the boundary. $\text{DT}(\mathbf{p})$ is pixel-dependent, varying with classification (foreground or background) and relative position. Pixels closer to the object's center receive higher values, while those farther away or in the background have lower values. We multiply the generated $\mathcal{P}_{\text{cdm}}$ and $\mathcal{P}_{\text{bdm}}$ with the original binary image $I$ to remove the background interference as

$$I_{\text{cdm}} = I \odot \mathcal{P}_{\text{cdm}}, \ I_{\text{bdm}} = I \odot \mathcal{P}_{\text{bdm}}, \tag{10}$$

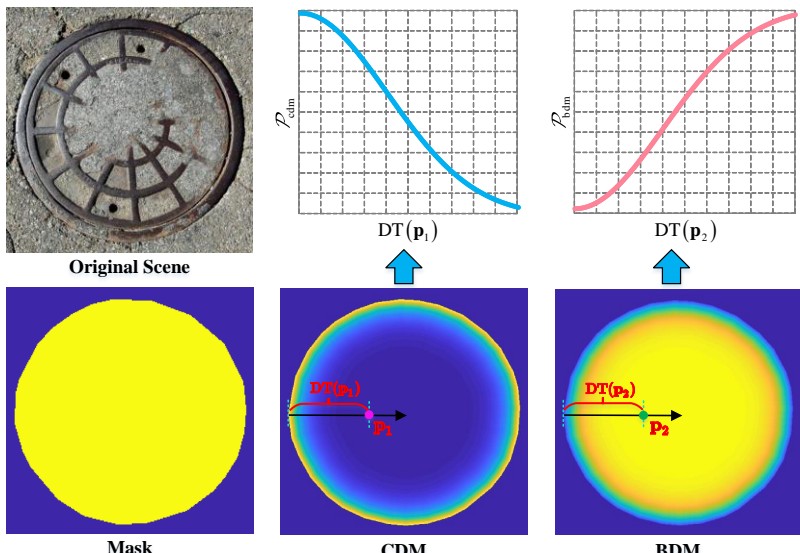

Figure 3: Example of label decoupling: the label is decomposed into a contour distribution map (CDM) and a body distribution map (BDM).

where $\odot$ represents element-wise multiplication. $I_{\text{cdm}}$ and $I_{\text{bdm}}$ mean the CDM and BDM as shown in Fig. 3, respectively. Accordingly, the segmentation task is decoupled into the inference subtasks for CDM and BDM.

**Body-Distribution Learning.** The body information is critical for determining the overall segmentation effect. We design a body-distribution learning branch to mine the accurate main region. We feed the multi-level side outputs of OFD into the designed contextual exploration (CE) module for extracting contextual information. Then, we introduce the hierarchical information fusion (HIF) module to aggregate the outputs from the multi-layer CE modules to obtain the BDM results. The details of CE and HIF are explained in Fig. 11 of Appendix A.3.

**Contour-Distribution Learning.** The contour is usually used as a prime cue to refine the object morphology for accurate segmentation. We design the contour-distribution learning branch consisting of the dual-path attention (DPA) to focus on learning contour information. The details of DPA are explained in Fig. 12 of Appendix A.4.

### 3.4 CONTOUR-BODY INTERACTION (CBI)

Considering the correlation between CDM and BDM, we propose the CBI in combination with graph convolutional neural networks. As shown in Fig. 4, the CBI consists of three main components: cross-layer correlation, polishing gate, and mask generation.

**Cross-layer Correlation.** For the input feature map $\{T_1, T_2\} \in \mathcal{R}^{C \times H \times W}$, we apply two $1 \times 1$ convolutional layers ($\mathcal{G}_{\text{edge}}$ and $\mathcal{G}_{\text{node}}$) to transform $\{T_1, T_2\}$ into two independent representations, and then extract the transformed feature patches into two groups, $i.e.$, $G' = \{\mathbf{p}'_i \mid 1 \leq i \leq K\}$ and $G'' = \{\mathbf{p}''_i \mid 1 \leq i \leq K\}$, via the unfolding operation $f_{\text{unfold}}$ (shown in Fig. 4). The feature patches in $G'$ and $G''$ have the following feature representations:

$$\mathbf{p}'_i = f_{\text{unfold}}\left(\mathcal{G}_{\text{edge}}\left(T_1\right)\right), \quad \mathbf{p}''_i = f_{\text{unfold}}\left(\mathcal{G}_{\text{node}}\left(T_2\right)\right), \tag{11}$$

where $G'$ is used to build graph connections and $G''$ is assigned as the graph nodes. Given a set of feature patches $G'$, we flatten each patch into a feature vector and compute feature similarity using the dot product, resulting in a similarity matrix $\mathbf{S} \in \mathcal{R}^{K \times K}$, defined as:

$$\mathbf{S} = \text{FC}\left(\text{Flatten}(G'_i)\right) \otimes \text{FC}\left(\text{Flatten}(G'_i)\right), \tag{12}$$

where $\otimes$ denotes the matrix multicaption. $\text{Flatten}\left(\cdot\right)$ is the flatten operator, $\text{FC}\left(\cdot\right)$ is the full connected layer. Consider $\mathbf{S}_{i,:}$, the $i$-th row of $\mathbf{S}$, representing the similarity of the $i$-th node to other nodes. We employ a dynamic number of neighbors for each node based on the nearest principle. This is achieved through a dynamic KNN (DKNN) module, which generates an adaptive threshold

Figure 4: Illustration of the CBI. It consists of three main components: cross-layer correlation, polishing gate, and mask generation.

for each node, selecting neighbors with similarities above this threshold as candidates. The average value of $\mathbf{S}_{i,:}$ represents the average importance of different nodes to the $i$-th node, denoted as $Q_i$. As shown in Fig. 4, to improve the adaptability, we apply the node-specific affine transformation to calculate $Q_i$ as:

$$Q_i = \frac{\varphi_2\left(p_i'\right)}{K} \sum_{k=1}^{K} \mathbf{S}_{i,k} + \varphi_1\left(p_i'\right) = \frac{\beta}{K} \sum_{k=1}^{K} \mathbf{S}_{i,k} + \alpha, \tag{13}$$

where $\alpha = \varphi_1\left(p_i'\right)$ and $\beta = \varphi_2\left(p_i'\right)$. $\varphi_1$ and $\varphi_2$ are two distinct $W_p \times H_p$ convolutional layers, embedding each node into specific affine transform parameters, $i.e.$, $\alpha$ and $\beta$. To achieve a different threshold truncation, we utilize the ReLU function to truncate input features and normalize the similarity of all connected nodes by the softmax function to calculate the attention weights by

$$\alpha_{i,j} = \frac{\exp\left(\mathbf{A}_{i,j}\right)}{\sum_{j \in \mathcal{N}_i} \exp\left(\mathbf{A}_{i,j}\right)}, j \in \mathcal{N}_i, \quad \mathbf{A}_{i,:} = \mathrm{ReLU}\left(\mathbf{S}_{i,:} - Q_i\right), \tag{14}$$

where $\mathbf{A} \in \mathcal{R}^{K \times K}$ is the adjacency matrix in which $\mathbf{A}_{i,j}$ is assigned the similarity weight if $\mathbf{p}_j'$ connect to $\mathbf{p}_i'$, otherwise equal to zero. $\mathcal{N}_i$ is the set of indexes of neighboring nodes. The feature aggregation process is a graph described as a weighted sum of all connected neighbors:

$$\hat{\mathbf{p}}_i = \sum_{j \in \mathcal{N}_i} \alpha_{i,j} \times \mathbf{p}_j'' = \sum_{j \in \mathcal{N}_i} \alpha_{i,j} \times \mathcal{G}_{\mathrm{node}}\left(\mathbf{p}_j\right). \tag{15}$$

We extract feature patches from the graph to aggregate into a feature map via folding operation. Overlapping regions are handled by averaging to suppress blocking effects. Global residual connectivity in the cross-layer correlation module further enhances the result. The output of this module is expressed as $r = f_{\mathrm{fold}}\left(\hat{\mathbf{p}}_1, \hat{\mathbf{p}}_2, ..., \hat{\mathbf{p}}_i, ..., \hat{\mathbf{p}}_K\right)$, $f_{\mathrm{fold}}\left(\cdot\right)$ is a folding operation, as shown in Fig. 4. The extracted cross-layer correlation matrix $r$ is normalized by a softmax operator along the rows and columns, respectively, to locate the object regions involved in the high-level semantics by

$$F_{\mathrm{corr}}^1 = \mathrm{Rp}\left(\mathrm{Rp}\left(\mathcal{G}_{\mathrm{edge}}\left(T_1\right)\right) \odot \mathcal{S}(r)\right), F_{\mathrm{corr}}^2 = \mathrm{Rp}\left(\mathrm{Rp}\left(\mathcal{G}_{\mathrm{node}}\left(T_2\right)\right) \odot \mathcal{S}\left(r^\top\right)\right), \tag{16}$$

where $\mathrm{Rp}(\cdot)$ is the reshape operation and $\mathcal{S}(\cdot)$ is the softmax operator. $F_{\mathrm{corr}}^1, F_{\mathrm{corr}}^2 \in \mathcal{R}^{C \times H \times W}$ are features containing rich location information. Since we perform matrix-based cross-layer correlation operations on $F_{\mathrm{corr}}^1$ and $F_{\mathrm{corr}}^2$ with low computational cost.

**Polishing Gate**. To address redundancy in $\{F_{\mathrm{corr}}^1, F_{\mathrm{corr}}^2\}$, we introduce an effective gating mechanism to refine location information. Using a $1 \times 1$ convolution, we generate response maps in

$[0,1]^{1 \times H \times W}$ for $\{F_{\text{corr}}^1, F_{\text{corr}}^2\}$. These maps filter out redundant information by gating mechanism as $F_{\text{gate}}^1 = \text{Sigmoid}\left(\text{Conv}_{1 \times 1}\left(F_{\text{corr}}^1\right)\right) \odot F_{\text{corr}}^1, F_{\text{gate}}^2 = \text{Sigmoid}\left(\text{Conv}_{1 \times 1}\left(F_{\text{corr}}^2\right)\right) \odot F_{\text{corr}}^2$, $\left\{F_{\text{gate}}^1, F_{\text{gate}}^2\right\} \in \mathcal{R}^{C \times H \times W}$ are the polished features. $\text{Conv}_{1 \times 1}$ is the $1 \times 1$ convolution layer.

**Mask Generation**. Moreover, we adopt the residual connection to merge $F_{\text{gate}}^1$ and $T_1$ as well as $F_{\text{gate}}^2$ and $T_2$, respectively, resulting $\hat{F}_{\text{gate}}^1$ and $\hat{F}_{\text{gate}}^2$ by $\hat{F}_{\text{gate}}^1 = \text{DSConv}\left(F_{\text{gate}}^1 + T_1\right), \hat{F}_{\text{gate}}^2 = \text{DSConv}\left(F_{\text{gate}}^2 + T_2\right)$, $\text{DSConv}(\cdot)$ is the $3 \times 3$ depth-wise separable convolution layer. The generated $\hat{F}_{\text{gate}}^1$ and $\hat{F}_{\text{gate}}^2$ are fused to generate the segmenatation map by $P_{\text{seg}} = \text{Sigmoid}\left(\text{Conv}_{1 \times 1}\left(\text{DSConv}\left(\hat{F}_{\text{gate}}^1 \odot \hat{F}_{\text{gate}}^2\right)\right)\right)$. We completely extract location information from $\hat{F}_{\text{gate}}^1$ and $\hat{F}_{\text{gate}}^2$ to accurately determine the object regions.

## 3.5 LOSS FUNCTION

To well train the FDTDNet, we combine the weighted BCE loss $\ell_{\text{wBCE}}$ (Wei et al. (2020)) and weighted IoU loss $\ell_{\text{wIOU}}$ (Wei et al. (2020)), that is, $L_{\text{s}} = \ell_{\text{wBCE}} + \ell_{\text{wIOU}}$ to perform supervised learning on the CDM, BDM, and final segmentation maps. Thus the total loss function is:

$$L_{\text{All}} = L_{\text{s}}(P_{\text{CDM}}, G_{\text{CDM}}) + L_{\text{s}}(P_{\text{BDM}}, G_{\text{BDM}}) + L_{\text{s}}(P_{\text{seg}}, G_{\text{seg}}), \tag{17}$$

where $P_{\text{CDM}}, P_{\text{BDM}}$, and $P_{\text{seg}}$ are the predicted CDM, BDM, and final segmentation maps, respectively. $G_{\text{CDM}}, G_{\text{BDM}}$, and $G_{\text{seg}}$ are the true CDM, BDM, and segmentation maps, respectively.

# 4 EXPERIMENTS

## 4.1 SETUPS

**Datasets.** We use the datasets (Yin et al. (2024)) for lensless object segmentation named directly captured (DIRC) dataset and display captured (DISC) dataset (Yin et al. (2024)). The DIRC dataset is used for testing, and it consists of 30 natural scene images directly captured from 10 different scenes. The DISC dataset is collected from Display, including 5.2K paired data for training (DISC-Train) and 0.7K for testing (DISC-Test). Note that the measurements are captured by FlatCam.

**Evaluation Metrics**. To quantitatively evaluate the performance of each method, we use six evaluation matrices, including mean absolute error ($\mathcal{M}$), mean E-measure ($E_\xi$) (Fan et al. (2021)), weighted F-measure($F_\beta^w$) (Margolin et al. (2014)), S-measure ($S_\alpha$) (Fan et al. (2017)), mean Dice (mDice), and mean IoU (mIoU).

**Implementation Details.** In our FDTDNet, the PVT pre-trained on ImageNet initializes the backbone. We train the FDTDNet by the Adam optimizer with "cosine" learning rate policy as $lr = 0.5 \times init\_r \times (1 + \cos(\pi * epoch/max\_epoch))$, where the initial learning rate $init\_r$ is set to $5 \times 10^{-4}$ and training epoch $epoch \in [1, max\_epoch]$, $max\_epoch = 100$. The whole network is trained with a batch size of $8$. All experiments are implemented in Pytorch 1.8.0 and trained on a Linux 20.04 server with a single GPU of NVIDIA RTX 3090.

**Compared Methods.** For a fair evaluation, we compare our method with following methods: (1) Current advanced object segmentation methods, including CDMNet (Song et al. (2023)), SINetV2 (Fan et al. (2022)), C2FNet (Chen et al. (2022)), OCENet (Liu et al. (2022)), Zoom-Net (Pang et al. (2022)), TransUnet (Chen et al. (2021)), and BDG-Net (Qiu et al. (2022)); And (2) Existing object inference methods for lensless imaging: LLI_T (Pan et al. (2021a)), Raw3dNet (Zhang et al. (2022)), EyeCoD (You et al. (2022)), LOINet (Yin et al. (2022)), and RecSegNet (Yin et al. (2024)). We employ open-source codes from public repositories to implement established comparison methods. To ensure consistency, all methods are retrained on a shared training dataset.

## 4.2 COMPARISON WITH STATE-OF-THE-ARTS

**Evaluation on DISC-Test Dataset.** Figure 5 displays segmentation results from our FDTDNet and various state-of-the-art methods (CDMNet, C2FNet, SINetV2, BDG-Net, OCENet, TransUnet, and

Table 1: Comparison of our FDTDNet and other 12 state-of-the-art methods. ↑ means that the more prominent, the better, and ↓ means that the more minor, the more remarkable. The first, second, and third-ranked performances are highlighted in **red**, **green**, and **blue**, respectively.

| Methods | DISC-Test | | | | | | DIRC | | | | | |
|---|---|---|---|---|---|---|---|---|---|---|---|---|
| | $F_\beta^w \uparrow$ | $\mathcal{M} \downarrow$ | $E_\xi \uparrow$ | $S_\alpha \uparrow$ | mDice ↑ | mIoU ↑ | $F_\beta^w \uparrow$ | $\mathcal{M} \downarrow$ | $E_\xi \uparrow$ | $S_\alpha \uparrow$ | mDice ↑ | mIoU ↑ |
| CDMNet | 0.535 | 0.241 | 0.688 | 0.652 | 0.618 | 0.473 | 0.739 | 0.117 | 0.805 | 0.738 | 0.756 | 0.679 |
| C2FNet | 0.493 | 0.291 | 0.639 | 0.562 | 0.557 | 0.405 | 0.713 | 0.119 | 0.793 | 0.763 | 0.826 | 0.704 |
| SINetV2 | 0.363 | 0.360 | 0.502 | 0.357 | 0.365 | 0.399 | 0.658 | 0.126 | 0.754 | 0.732 | 0.785 | 0.697 |
| BDG-Net | 0.508 | 0.261 | 0.665 | 0.582 | 0.553 | 0.405 | 0.645 | 0.151 | 0.746 | 0.714 | 0.768 | 0.679 |
| OCENet | 0.585 | 0.222 | 0.711 | 0.628 | 0.632 | 0.499 | 0.767 | 0.116 | 0.829 | 0.794 | 0.835 | 0.726 |
| TransUNet | 0.551 | 0.242 | 0.743 | 0.678 | 0.593 | 0.463 | 0.764 | 0.117 | 0.817 | 0.781 | 0.833 | 0.708 |
| ZoomNet | 0.661 | 0.177 | 0.811 | 0.753 | 0.716 | 0.605 | 0.773 | 0.115 | 0.815 | 0.787 | 0.840 | 0.752 |
| LLI_T | 0.721 | 0.137 | 0.802 | 0.748 | 0.764 | 0.669 | 0.742 | 0.115 | 0.821 | 0.759 | 0.817 | 0.732 |
| Raw3dNet | 0.749 | 0.118 | 0.827 | 0.752 | 0.777 | 0.674 | 0.779 | 0.105 | 0.834 | 0.778 | 0.836 | 0.749 |
| EyeCoD | 0.755 | 0.127 | 0.808 | 0.756 | 0.782 | 0.679 | 0.785 | 0.097 | 0.838 | 0.786 | 0.833 | 0.752 |
| LOINet | **0.763** | **0.129** | **0.832** | **0.764** | **0.799** | **0712** | **0.791** | **0.103** | **0.844** | **0.792** | **0.858** | **0.779** |
| RecSegNet | **0.866** | **0.067** | **0.907** | **0.861** | **0.879** | **0.818** | **0.854** | **0.078** | **0.858** | **0.891** | **0.867** | **0.824** |
| **FDTDNet** | **0.902** | **0.056** | **0.916** | **0.875** | **0.902** | **0.841** | **0.918** | **0.047** | **0.923** | **0.903** | **0.907** | **0.874** |

ZoomNet) on the DISC-Test dataset. Many comparison methods struggle with low-contrast (3rd, 4th, 6th rows) and cluttered backgrounds (1st, 2nd, 5th rows), failing to segment objects accurately due to their limited capacity to extract details from lensless measurements. In contrast, FDTDNet employs feature demultiplexing, yielding superior segmentation. Further analysis in Fig. 13 in Appendix A.6 compares our method with existing lensless segmentation techniques (LLI_T, Raw3dNet, EyeCoD, LOINet, and RecSegNet). While these methods perform well, FDTDNet achieves results closest to ground truths. Tab. 1 quantifies lensless object segmentation performance, showing FDTDNet outperforms all competitors across metrics. Specifically, it reduces $\mathcal{M}$ by 12.5% and improves $F_\beta^w$, $E_\xi$, $S_\alpha$, mDice, and mIoU by 3.3%, 0.4%, 2.0%, 2.0%, and 2.8%, respectively, compared to RecSegNet. Note that our method advances task decomposition by using CDM and BDM with a CBI module for mutual learning, tailored to lensless imaging's ambiguous boundaries. Unlike CDMNet's edge-based focus, our dual-branch design achieves more comprehensive segmentation.

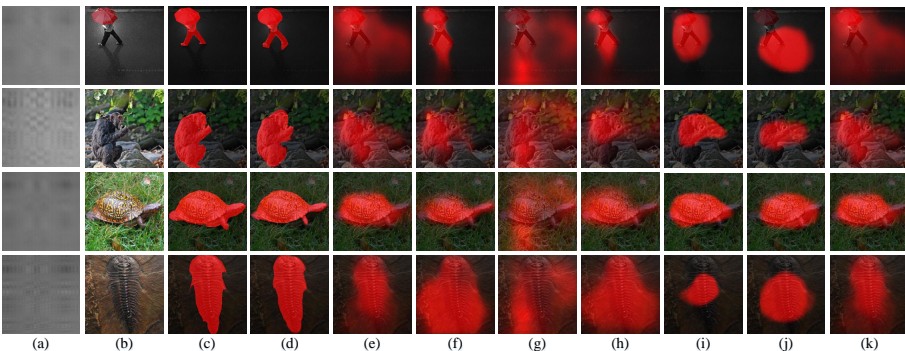

Figure 5: Comparison with state-of-the-art methods on the DISC-Test dataset. The (a) is the lensless measurements corresponding to real images (b); The (c) is the real segmentation maps corresponding to (b); The (d)–(k) are the segmentation results by our FDTDNet, CDMNet, C2FNet, SINetV2, BDG-Net, OCENet, TransUnet, and ZoomNet.

**Evaluation on DIRC Dataset.** With limited visual input causing failures in most comparison methods on DIRC dataset, we adopt the training setups in (Yin et al. (2024)) to obtain results, as shown in Tab. 1. While comparison methods with the setups in (Yin et al. (2024)) perform well on DIRC dataset due to simpler objects and uniform backgrounds, our FDTDNet consistently outperforms them across all metrics. Notably, it reduces $\mathcal{M}$ by 34.7% and enhances $F_\beta^w$, $E_\xi$, $S_\alpha$, mDice, and mIoU by 7.1%, 6.7%, 0.9%, 4.0%, and 5.3%, respectively, compared to RecSegNet. Fig. 6 show-

cases segmentation results, revealing that many methods struggle with accurate segmentation due to the lack of visual semantics in lensless measurements. In contrast, FDTDNet excels, demonstrating its robust generalization capability.

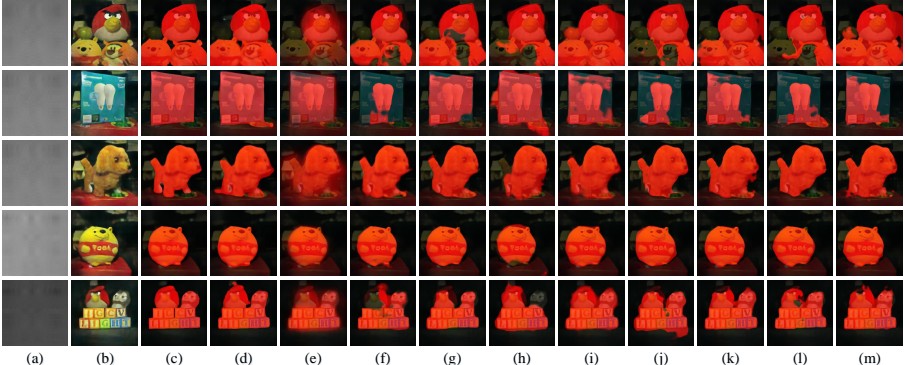

(a)  (b)  (c)  (d)  (e)  (f)  (g)  (h)  (i)  (j)  (k)  (l)  (m)

Figure 6: Comparison with state-of-the-art methods on the DIRC dataset. The (a) is lensless measurements; (b) is the restored images by FlatNet (Salman et al. (2022)); The (c) is the real segmentation maps corresponding to (b); The (d)–(m) are the segmentation results by our FDTDNet, CDMNet, C2FNet, SINetV2, BDG-Net, OCENet, TransUnet, ZoomNet, LOINet, and RecSegNet.

### 4.3 ABLATION STUDIES

**Ablation Studies on Tasks.** Table 2 presents the comparison results obtained through various task supervisions. Note that "Segm" refers to the direct segmentation map supervision, "Edge" denotes the edge supervision, "CDM" represents the CDM supervision, and "BDM" symbolizes the BDM supervision. The configuration incorporating CDM outperforms the configuration involving edge, suggesting that CDM supervision is more effective than edge supervision. Furthermore, the amalgamation featuring BDM exhibits superior performance compared with the configuration incorporating segmentation maps. This validates that a more effective feature representation could be learned for the body regions without interfering with edges.

Table 2: Comparison of different task supervision on DISC-Test dataset. The first-ranked result is highlighted in **red**.

| Task | $F_\beta^w \uparrow$ | $\mathcal{M} \downarrow$ | $E_\xi \uparrow$ | $S_\alpha \uparrow$ | mDice $\uparrow$ | mIoU $\uparrow$ |
|---|---|---|---|---|---|---|
| CDM + Segm | 0.889 | 0.065 | 0.881 | 0.864 | 0.887 | 0.814 |
| Edge+ Segm | 0.882 | 0.066 | 0.882 | 0.861 | 0.885 | 0.815 |
| BDM + Edge | 0.885 | 0.066 | 0.885 | 0.861 | 0.883 | 0.814 |
| BDM + Segm | 0.889 | 0.067 | 0.886 | 0.863 | 0.884 | 0.814 |
| Full model | **0.902** | **0.056** | **0.916** | **0.875** | **0.902** | **0.841** |

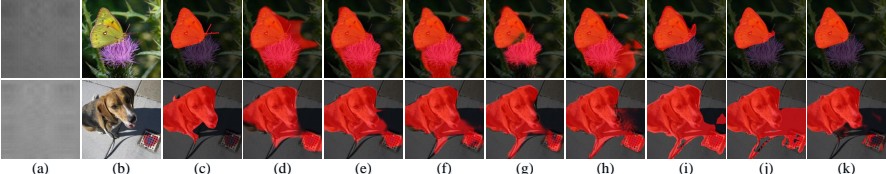

(a)  (b)  (c)  (d)  (e)  (f)  (g)  (h)  (i)  (j)  (k)

Figure 7: Ablation studies on the DISC-Test dataset. The (d)–(k) corresponding to $\mathrm{Conf}_1$–$\mathrm{Conf}_8$. The (a) is the lensless measurements of the underlying scenes (b) by the lensless camera; The (c) is the ground truth segmentation maps corresponding to (b).

**Ablation Studies on Components.** We explore the effectiveness of each component in our FDTD-Net. Note that removed HIF is replaced by upsampling + concatenation + convolution, removed DPA is replaced by $3 \times 3$ convolution layer, and removed CBI is replaced by concatenation. For the above configuration, we obtain the corresponding evaluation results, as illustrated in Fig. 7 and

Tab. 3. From the results in Fig. 7 $(d)$–$(h)$ and $\mathrm{Conf}_1$–$\mathrm{Conf}_5$ in Tab. 3, the removal of each component (*i.e.*, CE, OFD, DPA, HIF, and CBI) results in a drop of segmentation performance. These results demonstrate the effectiveness of the individual components.

**Ablation Studies on Loss Functions.** To further explore the effect of our method, we analyze each loss function, and the corresponding results are shown in Fig. 7(i), (j), as well as $\mathrm{Conf}_6$ and $\mathrm{Conf}_7$ in Tab. 3. Employing only either $\ell_{\mathrm{wBCE}}$ or $\ell_{\mathrm{wIOU}}$ leads to a degradation of predicted effect, while better results are obtained by training our network with the total loss function (*i.e.*, Fig. 7(k) and $\mathrm{Conf}_8$ in Tab. 3). These results indicate that a tailored loss functions are necessary for our FDTDNet.

Table 3: Ablation studies on DISC-Test dataset. The first-ranked result is highlighted in **red**.

| $\mathrm{Conf}_{\mathrm{ID}}$ | Component | | | | | Loss Function | | Evaluation Metrics | | | | | |
|---|---|---|---|---|---|---|---|---|---|---|---|---|---|
| | OFD | CE | DPA | HIF | CBI | $L_{\mathrm{wIOU}}$ | $L_{\mathrm{wBCE}}$ | $F_\beta^w \uparrow$ | $\mathcal{M} \downarrow$ | $E_\xi \uparrow$ | $S_\alpha \uparrow$ | mDice $\uparrow$ | mIoU $\uparrow$ |
| $\mathrm{Conf}_1$ | | ✓ | ✓ | ✓ | ✓ | ✓ | ✓ | 0.652 | 0.231 | 0.694 | 0.667 | 0.708 | 0.588 |
| $\mathrm{Conf}_2$ | ✓ | | ✓ | ✓ | ✓ | ✓ | ✓ | 0.676 | 0.207 | 0.726 | 0.723 | 0.771 | 0.613 |
| $\mathrm{Conf}_3$ | ✓ | ✓ | | ✓ | ✓ | ✓ | ✓ | 0.683 | 0.189 | 0.747 | 0.726 | 0.805 | 0.621 |
| $\mathrm{Conf}_4$ | ✓ | ✓ | ✓ | | ✓ | ✓ | ✓ | 0.852 | 0.084 | 0.867 | 0.805 | 0.877 | 0.805 |
| $\mathrm{Conf}_5$ | ✓ | ✓ | ✓ | ✓ | | ✓ | ✓ | 0.746 | 0.173 | 0.776 | 0.728 | 0.748 | 0.698 |
| $\mathrm{Conf}_6$ | ✓ | ✓ | ✓ | ✓ | ✓ | ✓ | | 0.824 | 0.103 | 0.857 | 0.804 | 0.829 | 0.783 |
| $\mathrm{Conf}_7$ | ✓ | ✓ | ✓ | ✓ | ✓ | | ✓ | 0.867 | 0.062 | 0.901 | 0.837 | 0.878 | 0.803 |
| $\mathrm{Conf}_8$ | ✓ | ✓ | ✓ | ✓ | ✓ | ✓ | ✓ | **0.902** | **0.056** | **0.916** | **0.875** | **0.902** | **0.841** |

## 4.4 LIMITATIONS

While our method performs well in conventional scenarios, it shows performance degradation in unconventional cases, as analyzed in Fig. 8. Specifically, non-uniform or hollow target regions lead to missed detections (first row), small targets result in false positives (second row), and blurred target boundaries cause significant false positives. These issues arise from inherent challenges in lensless imaging, such as optical cross-talk and complex scenes. The method excels with flat, high-intensity, large targets but struggles in more complex conditions. Future work should address these limitations by (1) expanding datasets to include diverse scenarios, (2) applying domain adaptation for improved generalization, and (3) adopting frequency-adaptive techniques to mitigate cross-talk artifacts.

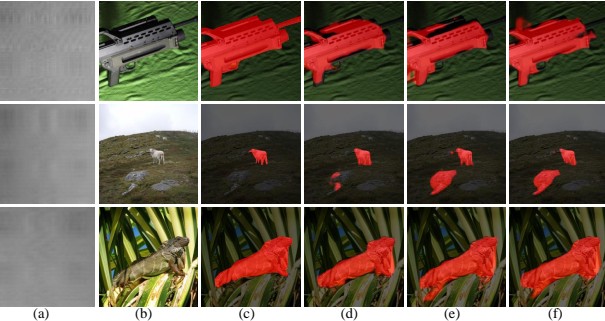

Figure 8: Illustration of failure cases. The (a) is the lensless measurements of the underlying scene (b) by the lensless camera; The (c) is the ground truth segmentation maps corresponding to (b); The (d)–(f) are the segmentation results from our method, LOINet, and RecSegNet, respectively.

## 5 CONCLUSION

This paper addresses the challenges of lensless object segmentation by the proposed one-step method, FDTDNet, developing an optical-aware feature demultiplexing mechanism and decomposing the task into CDM and BDM inference subtasks. For the former, the FDTDNet applies a new extractor combining OFD and PVT for reconstructing semantic features. Moreover, for the latter, we enhance lensless object segmentation performance by incorporating contour-distribution and body-distribution learning branches and a contour-body interaction strategy. Extensive experiments on the DISC-Test and DIRC datasets show that our FDTDNet outperforms state-of-the-art methods across various evaluation metrics and highlights its potential in advancing the field of lensless imaging.

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

## A APPENDIX

### A.1 THE DERIVATION DETAILS OF THE OFD

Combining Eq. (1) with the lensless imaging model in Eq. (2), the above semantic features $Y_{\theta_i}$ ($i = 1, 2, 3, 4$) is modeled as:

$$Y_{\theta_i} = A_{\mathrm{L},\theta_i} X_{\theta_i} A_{\mathrm{R},\theta_i}^\top + \xi, \tag{18}$$

where $X_{\theta_i}$, $A_{L,\theta_i}$, and $A_{R,\theta_i}$ denote the $X$, $A_L$, and $A_R$ in the feature space. Therefore, the task of reasoning about $X_{\theta_i}$ from $Y_{\theta_i}$ can be modeled as an inverse problem. To obtain $X_{\theta_i}$ for boosting the lensless object segmentation task, inspired by (Salman et al. (2022)), our OFD-based extractor is designed as the Tikhonov regularization problem as:

$$\arg\min_{X_{\theta_i}} \left\| Y_{\theta_i} - A_{L,\theta_i} X_{\theta_i} A_{R,\theta_i}^\top \right\|_2^2 + K_{\theta_i} \left\| X_{\theta_i} \right\|_2^2, \tag{19}$$

where $K_\theta$ is the learnable regularization parameter.

The Eq. (19) represents a convex optimization problem, implying the existence of a unique minimum, which corresponds to the function value at the point where its derivative equals zero. To solve this, we set the derivative of Eq. (19) to zero, yielding:

$$A_{L,\theta_i}^\top (A_{L,\theta_i} X_{\theta_i} A_{R,\theta_i}^\top - Y_{\theta_i}) A_{R,\theta_i} + K_{\theta_i} X_{\theta_i} = 0. \tag{20}$$

Expanding the first term and rearranging yields:

$$A_{L,\theta_i}^\top A_{L,\theta_i} X_{\theta_i} A_{R,\theta_i}^\top A_{R,\theta_i} + K_{\theta_i} X_{\theta_i} = A_{L,\theta_i}^\top Y_{\theta_i} A_{R,\theta_i}. \tag{21}$$

Getting the SVD of $A_{L,\theta_i}$ and $A_{R,\theta_i}$ as:

$$\begin{aligned}
A_{L,\theta_i} &\overset{\text{SVD}}{=} U_{L,\theta_i} \Sigma_{L,\theta_i} V_{L,\theta_i}^\top, & A_{R,\theta_i} &\overset{\text{SVD}}{=} U_{R,\theta_i} \Sigma_{R,\theta_i} V_{R,\theta_i}^\top \\
A_{L,\theta_i}^\top &\overset{\text{SVD}}{=} V_{L,\theta_i} \Sigma_{L,\theta_i}^\top U_{L,\theta_i}^\top, & A_{R,\theta_i}^\top &\overset{\text{SVD}}{=} V_{R,\theta_i} \Sigma_{R,\theta_i}^\top U_{R,\theta_i}^\top.
\end{aligned} \tag{22}$$

Thus we can further obatain:

$$\begin{aligned}
A_{L,\theta_i}^\top A_{L,\theta_i} &\overset{\text{SVD}}{=} V_{L,\theta_i} \Sigma_{L,\theta_i}^\top U_{\top,\theta_i}^\top U_{L,\theta_i} \Sigma_{L,\theta_i} V_{L,\theta_i}^\top = V_{L,\theta_i} \Sigma_{L,\theta_i}^2 V_{L,\theta_i}^\top \\
A_{R,\theta_i}^\top A_{R,\theta_i} &\overset{\text{SVD}}{=} V_{R,\theta_i} \Sigma_{R,\theta_i}^\top U_{R,\theta_i}^\top U_{R,\theta_i} \Sigma_{R,\theta_i} V_{R,\theta_i}^\top = V_{R,\theta_i} \Sigma_{R,\theta_i}^2 V_{R,\theta_i}^\top.
\end{aligned} \tag{23}$$

Combining Eq. (23) with Eq. (23), we can obtain:

$$V_{L,\theta_i} \Sigma_{L,\theta_i}^2 V_{L,\theta_i}^\top X_{\theta_i} V_{R,\theta_i} \Sigma_{R,\theta_i}^2 V_{R,\theta_i}^\top + K_{\theta_i} X_{\theta_i} = V_{L,\theta_i} \Sigma_{L,\theta_i} U_{L,\theta_i}^\top Y_{\theta_i} U_{R,\theta_i} \Sigma_{R,\theta_i} V_{R,\theta_i}^\top. \tag{24}$$

Multiplying both sides of the Eq. (24) with $V_{L,\theta_i}^\top$ from the left and $V_{R,\theta_i}$ from the right yields:

$$\Sigma_{L,\theta_i}^2 V_{L,\theta_i}^\top X_{\theta_i} V_{R,\theta_i} \Sigma_{R,\theta_i}^2 + K_{\theta_i} V_{L,\theta_i}^\top X_{\theta_i} V_{R,\theta_i} = \Sigma_{L,\theta_i} U_{L,\theta_i}^\top Y_{\theta_i} U_{R,\theta_i} \Sigma_{R,\theta_i}. \tag{25}$$

Let $\sigma_{L,\theta_i}$ and $\sigma_{R,\theta_i}$ denote the diagonal entries of $\Sigma_{L,\theta_i}^2$ and $\Sigma_{R,\theta_i}^2$, respectively, yields:

$$V_{L,\theta_i}^\top X_{\theta_i} V_{R,\theta_i} \odot (\sigma_{L,\theta_i} \sigma_{R,\theta_i}^\top) + K_{\theta_i} V_{L,\theta_i}^\top X_{\theta_i} V_{R,\theta_i} = \Sigma_{L,\theta_i} U_{L,\theta_i}^\top Y_{\theta_i} U_{R,\theta_i} \Sigma_{R,\theta_i}, \tag{26}$$

where $\odot$ denotes element-wise multiplication. We further obtain

$$V_{L,\theta_i}^\top X_{\theta_i} V_{R,\theta_i} = (\Sigma_{L,\theta_i} U_{L,\theta_i}^\top Y_{\theta_i} U_{R,\theta_i} \Sigma_{R,\theta_i}) ./ (\sigma_{L,\theta_i} \sigma_{R,\theta_i}^\top + K_{\theta_i}), \tag{27}$$

where ./ denotes element-wise division. Therefore, the solution of Eq. (19) is written as

$$\begin{aligned}
\hat{X}_{\theta_i} &= \text{OFD}(Y_{\theta_i}; A_{L,\theta_i}, A_{R,\theta_i}) \\
&= V_{L,\theta_i} [(\Sigma_{L,\theta_i} U_{L,\theta_i}^\top Y_{\theta_i} U_{R,\theta_i} \Sigma_{R,\theta_i}) ./ (\sigma_{L,\theta_i} \sigma_{R,\theta_i}^\top + K_{\theta_i})] V_{R,\theta_i}^\top,
\end{aligned} \tag{28}$$

where the $A_{L,\theta_i}$ and $A_{R,\theta_i}$ are updated by $A_{L,\theta_i} = f_{A_L}(A_L)$ and $A_{R,\theta_i} = f_{A_R}(A_R)$. $f_{A_L}(\cdot)$ and $f_{A_R}(\cdot)$ represent the $3 \times 3$ convolution layer + batch normalization (BN) + ReLU + down-sampling operator. We denote each side output as $X_{\theta_i}$ for $\hat{X}_{\theta_i}$. Note that $A_L$ and $A_R$ are primarily associated with the system's system function and do not inherently contain scene-specific information, limiting their semantic content. With simpler convolutional operations for $A_L$ and $A_R$, network complexity is reduced while maintaining efficiency. Figure 9 shows the output results at different levels of OFD. As seen, OFD focuses on deriving semantically relevant features, such as object contours, to drive downstream tasks, rather than reconstructing visual details. This approach effectively prevents the leakage of visual information.

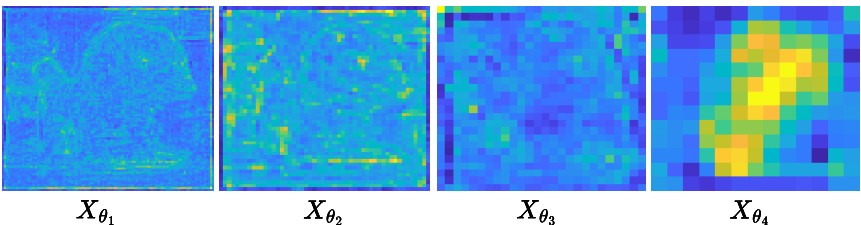

$X_{\theta_1}$ $\qquad$ $X_{\theta_2}$ $\qquad$ $X_{\theta_3}$ $\qquad$ $X_{\theta_4}$

Figure 9: Examples of output results from different levels of OFD. All results are zoomed to the same visualization size for comparison.

## A.2 THE DETAILS OF DATASETS

To perform object segmentation tasks for lensless imaging measurements, we construct two datasets named directly captured (DIRC) dataset and display captured (DISC) dataset. DIRC dataset is a testing dataset by directly capturing natural scenes containing 30 images across 10 scenes. DISC dataset is collected from Display-Captured dataset[1] (Khan et al. (2019)) containing 1000 categories of scenarios and the corresponding lensless imaging measurements. By removing unqualified scenes, we obtain 5.9K paired images with 869 categories, which cover flying, aquatic, terrestrial, amphibians, sky, vegetation, and indoor categories. Each category has at least 1 scenario and at most 10 scenarios. The DISC dataset includes 5.2K paired data for training (called DISC-Train) and 0.7K paired data for testing (called DISC-Test). The construction steps of these two datasets are detailed as follows.

First, we use $Eiseg^2$ software (a well-known datasets annotation application) combined with manual refinement to label binary maps $I_{\mathrm{mask}}$ for the two datasets.

Next, to perform the multi-task learning strategy with body distribution maps (BDM) $I_{\mathrm{bdm}}$ and contour distribution maps (CDM) $I_{\mathrm{cdm}}$, we acquire $I_{\mathrm{bdm}}$ and $I_{\mathrm{cdm}}$ via Eqs. (9) and (10) of the main manuscript.

Finally, we perform a double-check to ensure the accuracy of the labels, $i.e.$, $I_{\mathrm{mask}}$, $I_{\mathrm{bdm}}$, and $I_{\mathrm{cdm}}$.

Fig. 10 presents some examples showing the reliable annotation of our datasets. Note that the DISC-Train dataset is used to train both our method and the baselines, while the DISC-Test and DIRC datasets are employed for testing to evaluate the performance of each method.

## A.3 THE DETAILS OF BODY-DISTRIBUTION LEARNING

The body information is critical for determining the overall segmentation effect. Thus, we design a body-distribution learning branch consisting of three contextual exploration (CE) modules and a hierarchical information fusion (HIF) module.

**Contextual Exploration (CE).** As illustrated in Xia et al. (2024), for the human eye, group receptive fields of different sizes are beneficial for enhancing the perception of tiny areas near the focal point of the retina. Following this strategy, we propose a CE module that simulates the mechanism of the human eye in perceiving external objects to obtain a coarse representation of their bodies. The CE module consists of five branches, denoted as $b_k$ ($k = 1, 2, ..., 5$), as shown in Fig. 11. Except for the first and last branches with only one $1 \times 1$ convolution, other branches have four convolutions with a size of $1 \times 1$, $1 \times (2k-1)$, $(2k-1) \times 1$, and $3 \times 3$. First, the outputs of the first four branches are combined by concatenation, followed by a convolution to adjust the channel number to match that of $b_5$. Then, the results above are element-wise multiplied with the output of $b_5$ and then fed into the ReLU activation function to obtain the final features. As shown in Fig. 2, we cascade the CE module at the end of the OFD to get the features $C_2$, $C_3$, and $C_4$, respectively.

**Hierarchical Information Fusion (HIF).** We aggregate three outputs of the CE modules, $i.e.$, $C_2$, $C_3$, and $C_4$, to refine the object regions embedded. Unlike the way the partial decoder works, the

---
[1] https://siddiquesalman.github.io/flatnet/
[2] https://github.com/PaddleCV-SIG/EISeg

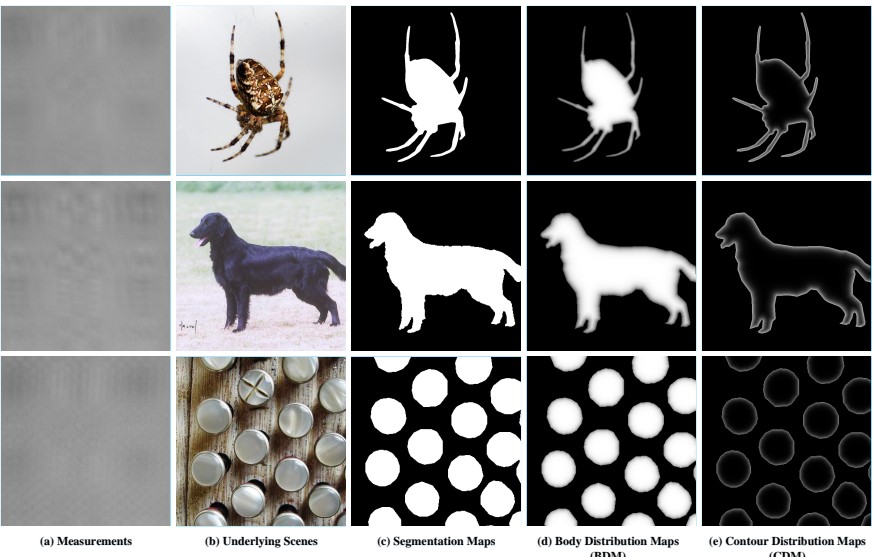

(a) Measurements  (b) Underlying Scenes  (c) Segmentation Maps  (d) Body Distribution Maps (BDM)  (e) Contour Distribution Maps (CDM)

Figure 10: Examples of our dataset. (a)–(e) represent lensless imaging measurements, underlying scenes, ground truth (GT), BDM, and CDM.

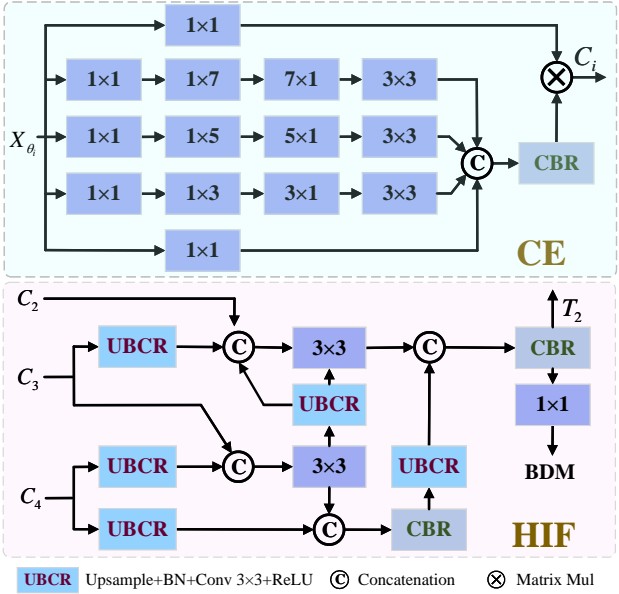

UBCR Upsample+BN+Conv 3×3+ReLU  C Concatenation  ⊗ Matrix Mul

Figure 11: Illustration of CE and HIF.

HIF modifies the skip between different scale features and neighborhood features to sufficiently enhance the bodies of objects and compensate for the details. The detailed structure is shown in Fig. 11. The HIF outputs $T_2$, fed into one $1 \times 1$ convolution layer for obtaining the BDM.

### A.4    THE DETAILS OF CONTOUR-DISTRIBUTION LEARNING

The contour information is usually used as a prime cue to refine the object morphology for accurate segmentation. We design the contour-distribution learning branch consisting of the DPA to focus on learning contour information. As shown in Fig. 12, through a $3 \times 3$ convolution layer, we first transform the output of the first OFD, $i.e.$, $X_{\theta_1} \in \mathcal{R}^{3 \times H \times W}$, into $F_1 \in \mathcal{R}^{C \times H \times W}$, where $C$,

Table 4: Ablation study on the weight setting of loss functions.

| ID | Configuration | $F_\beta^w \uparrow$ | $\mathcal{M} \downarrow$ | $E_x i \uparrow$ | $S_\alpha \uparrow$ | mDice $\uparrow$ | mIoU $\uparrow$ |
|---|---|---|---|---|---|---|---|
| #1 | $L_{\text{wIOU}} + 0.5 * L_{\text{wBCE}}$ | 0.898 | 0.056 | 0.913 | 0.872 | 0.899 | 0.839 |
| #2 | $0.5 * L_{\text{wIOU}} + L_{\text{wBCE}}$ | 0.893 | 0.057 | 0.909 | 0.864 | 0.892 | 0.831 |
| #3 | $0.5 * L_{\text{wIOU}} + 0.5 * L_{\text{wBCE}}$ | 0.899 | 0.056 | 0.911 | 0.873 | 0.898 | 0.837 |
| #4 | $L_{\text{wBCE}}$ | 0.867 | 0.062 | 0.901 | 0.837 | 0.878 | 0.803 |
| #5 | $L_{\text{wIOU}}$ | 0.824 | 0.103 | 0.857 | 0.804 | 0.829 | 0.783 |
| #6 | $L_{\text{wIOU}} + L_{\text{wBCE}}$ | 0.902 | 0.056 | 0.916 | 0.875 | 0.902 | 0.841 |

$H$, and $W$ are the channel, height, and width of $F_1$, respectively. Then, we apply global average pooling (GAP) in the spatial dimension of $F_1$ to calculate channel-wise statistics and a channel downscaling convolution to generate a feature representation. Further, the feature representation is passed through two parallel channel-upscaling convolutions to generate two feature descriptors, *i.e.*, $V_1$ and $V_2$, each of dimension is $C \times 1 \times 1$. Moreover, attentional activations $W_1$ and $W_2$ after softmax of $V_1$ and $V_2$ are generated for calibration and aggregation of $F_1$. Finally, we add up the two results obtained to get $T_1$.

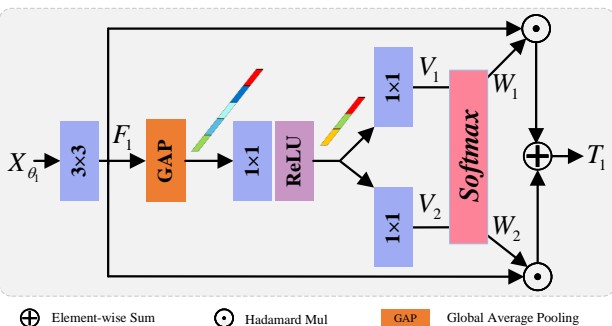

Figure 12: Illustration of the DPA.

### A.5 ABLATION STUDY ON THE COEFFICIENTS FOR LOSS FUNCTIONS

To provide a more comprehensive analysis of our method, we conducted additional experiments on the weight selection of each loss function, building upon the original ablation studies. The quantitative results are presented in Tab. 4. The results demonstrate that variations in performance across different weight configurations are marginal. Given above, we empirically adopted a 1:1 weight ratio.

### A.6 THE COMPARISON RESULTS BY OUR METHOD AND EXISTING LENSLESS SEGMENTATION TECHNIQUES

For fair evaluation, we also select state-of-the-art method (LLI_T, Raw3dNet, EyeCoD, LOINet, RecSegNet) for comparisons as shown in Fig. 13. Our method demonstrates more precise segmentation results compared to these methods.

### A.7 COMPLEXITY ANALYSIS

Fig. 14 displays the comparison results for complexity among the aforementioned 9 methods and our FDTDNet, considering parameters (Param), Floating Point Operations (FLOPs), and Frame Per Second (FPS). Our method features 25.87M parameters and 6.82G FLOPs, which are at an intermediate level. Furthermore, our method achieves a frame rate of 35.9 FPS, thereby fulfilling the essential real-time processing requirements. These results highlight that our FDTDNet achieves a favorable balance between performance and complexity.

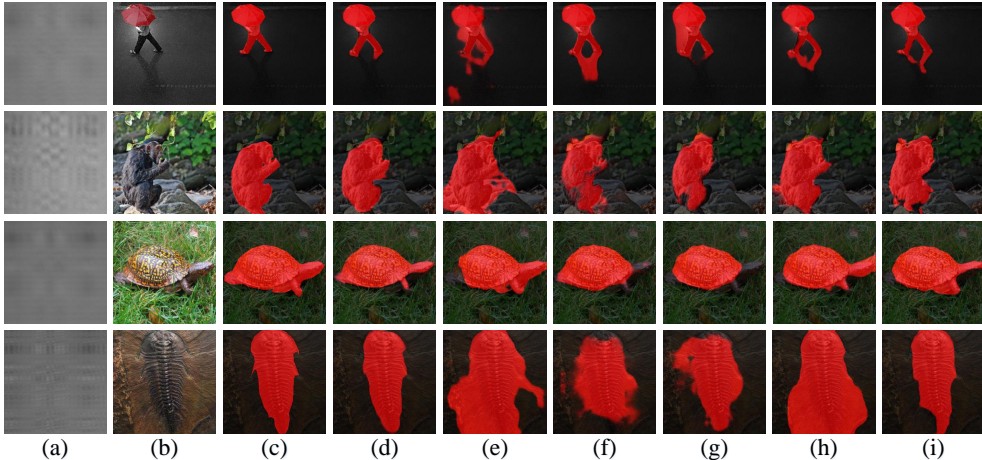

| (a) | (b) | (c) | (d) | (e) | (f) | (g) | (h) | (i) |

Figure 13: Comparison with state-of-the-art methods on the DISC-Test dataset. The (a) is the lens-less measurements corresponding to real images (b); The (c) is the real segmentation maps corresponding to (b); The (d)–(i) are the segmentation results by our FDTDNet, LLI_T, Raw3dNet, EyeCoD, LOINet, and RecSegNet.

### A.8 THE COMPARISON RESULTS BY OUR METHOD AND "RECONSTRUCTION + SEGMENTATION" TWO-STEP METHODS

While the focus of this paper is on the architecture and its potential benefits, we acknowledge the importance of comparing our method with traditional reconstruction-based methods. To this end, we employ FlatNet to reconstruct the underlying scene, followed by segmentation using methods such as CDMNet, BDG-Net, and ZoomNet as the "reconstruction + segmentation" two-step methods. Our method, however, retains its original configuration. The comparative results shown in Fig. 15 clearly demonstrate that our method outperforms these state-of-the-art methods in segmentation accuracy.

### A.9 MULTI-OBJECT SEGMENTATION RESULTS BY COMPARISON METHODS AND OURS

To further demonstrate the potential of our method in multi-object segmentation, we provide additional visualization results, as shown in Fig. 16. The figure illustrates that while all methods achieve partial segmentation of multiple objects, false positives and missed detections increase as the number of objects grows. Compared to other methods, ours achieves significantly higher segmentation accuracy. However, we acknowledge that there is still room for improvement in multi-object segmentation performance. In future work, we aim to enhance this aspect to increase its practical applicability.

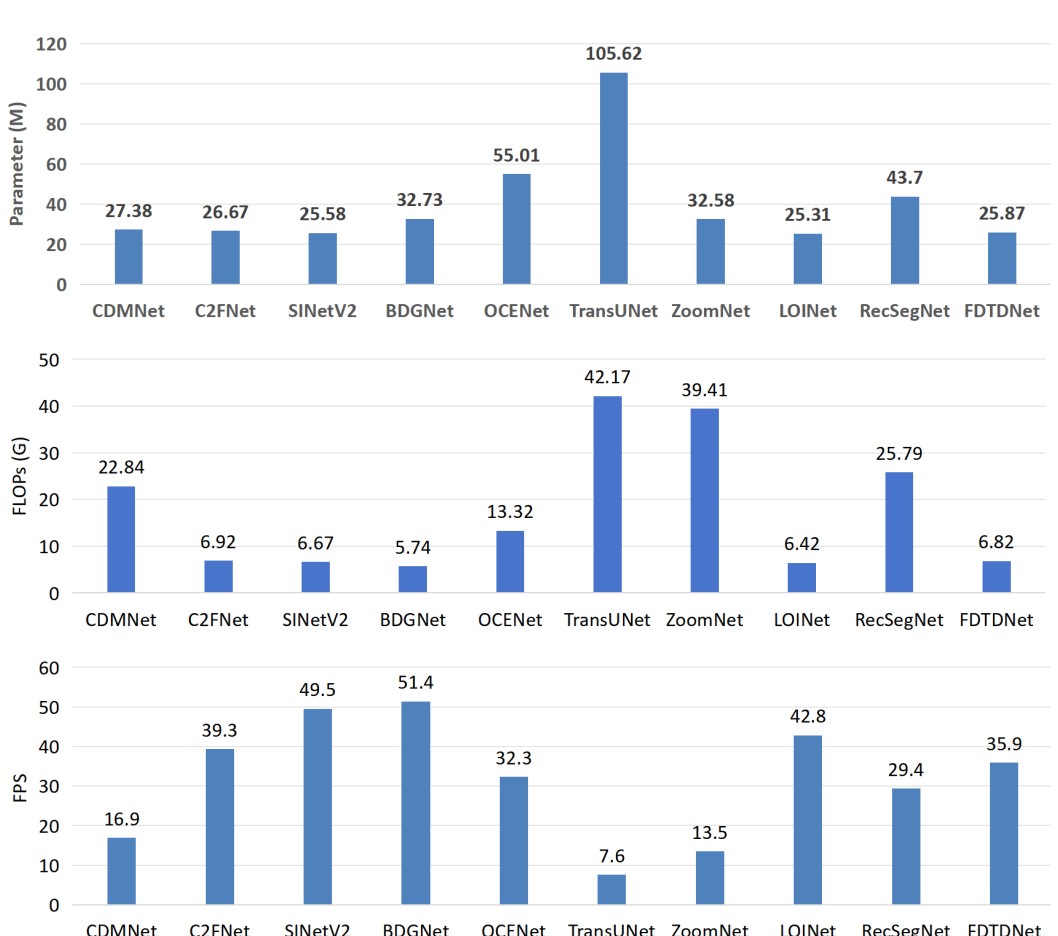

Figure 14: Complexity analysis of our FDTDNet and other state-of-the-art methods in terms of Param, FLOPs and FPS.

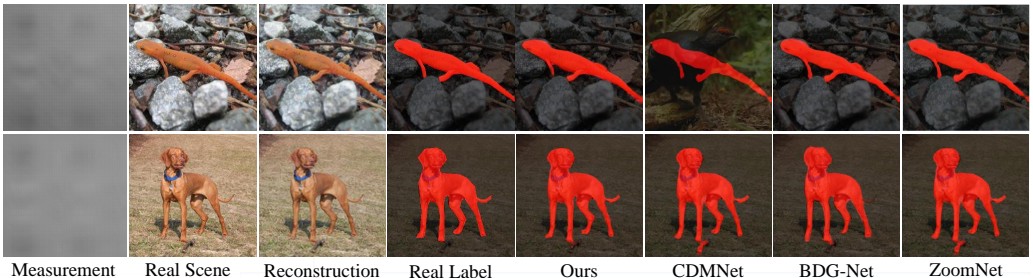

Figure 15: Comparison experiment between the "reconstruction + segmentation" two-step method and ours.

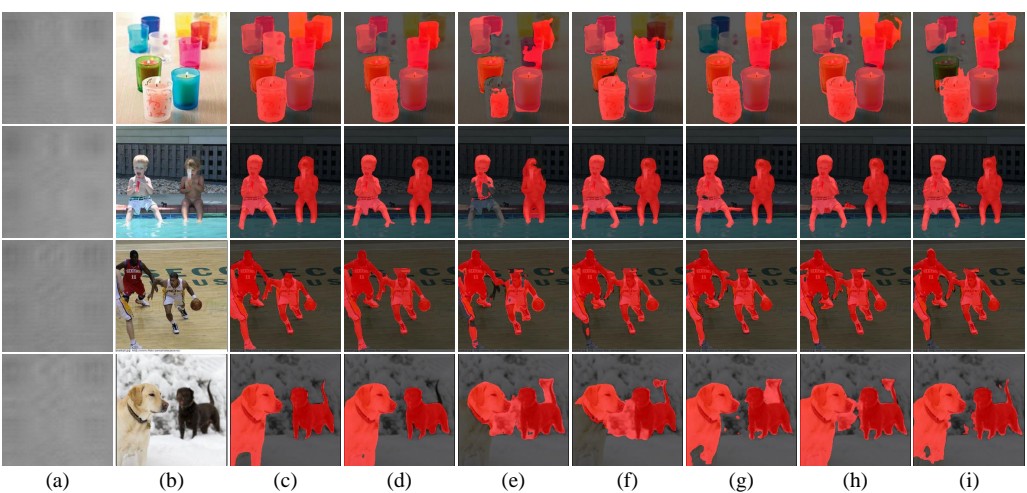

(a) (b) (c) (d) (e) (f) (g) (h) (i)

Figure 16: Multi-object segmentation results by comparison methods and ours. The (a) is the lensless measurements corresponding to real images (b); The (c) is the real segmentation maps corresponding to (b); The (d)–(i) are the segmentation results by our FDTDNet, LLI_T, Raw3dNet, EyeCoD, LOINet, and RecSegNet.

