# OpenReview forum: "FDTDNet: Privacy-Preserving Lensless Object Segmentation via Feature Demultiplexing and Task Decoupling"
_ICLR.cc/2025/Conference — ICLR 2025 Conference Withdrawn Submission_

### Official Review · Reviewer_8fG3 · 2024-10-29

**Soundness:** 3
**Presentation:** 3
**Contribution:** 3
**Rating:** 8
**Confidence:** 4

**Summary:**

The authors propose a one-step method without intermediate image reconstruction, addressing privacy concerns and computational efficiency.

**Strengths:**

1. Introduces an Optical-Aware Feature Demultiplexing mechanism that enhances feature extraction from lensless measurements.
2. Effectively decouples segmentation into contour and body tasks, leveraging a mutual learning strategy.
3. Demonstrates superior performance on two datasets, outperforming state-of-the-art methods in multiple metrics.

**Weaknesses:**

1. The performance of the network is not analyzed, such as the number of parameters, number of floating-point operations, inference time, etc.
2. Lack of explanation and verification of the weight setting of the hybrid loss function.
3. The paper does not explain the advantages of this one-step segmentation over the prior visual reconstruction method, and the experiment does not compare it with another method.
4. There is a lack of a more detailed description of the datasets. According to my understanding, are these datasets all synthetic? Are the measurements of the images synthesized using prior knowledge?

**Questions:**

1. The performance of the network is not analyzed, such as the number of parameters, number of floating-point operations, inference time, etc.
2. Lack of explanation and verification of the weight setting of the hybrid loss function.
3. The paper does not explain the advantages of this one-step segmentation over the prior visual reconstruction method, and the experiment does not compare it with another method.
4. There is a lack of a more detailed description of the datasets. According to my understanding, are these datasets all synthetic? Are the measurements of the images synthesized using prior knowledge?

---

> ### Author Response · Authors · 2024-11-25
>
> We sincerely thank you for your valuable feedback and acknowledgment of our work. To ensure clarity and comprehensively address your concerns, we have systematically provided detailed responses to each comment.
>
> **Weaknesses:**
>
> **Q1：The performance of the network is not analyzed, such as the number of parameters, number of floating-point operations, inference time, etc.**
>
> We sincerely appreciate your concern regarding the computational complexity of our network. As Ablation study on the weight setting of loss functionsnoted in  **Appendix A.7** , we have provided the number of parameters and the number of floating-point operations for each method. Our method effectively balances segmentation performance with computational complexity. In response to your suggestion, we incorporate the inference time in the revised manuscript (Also in **Appendix A.7**) to further evaluate the performance of our method. Experimental results demonstrate that our method achieves a frame rate of **35.9 FPS**, ranking just behind BDG-Net and ZoomNet, thereby fulfilling the essential real-time processing requirements.
>
> **Q2: Lack of explanation and verification of the weight setting of the hybrid loss function.**
>
> We sincerely appreciate your valuable feedback regarding the weight setting of the hybrid loss function. According to your suggestion, we have included additional experiments and clarifications on the explanation and verification of the coefficients for the loss function in  **Appendix A.5** . Here, we provide a brief explanation. The weight setting in the hybrid loss function is designed to balance the contributions of each component (BCE loss and IoU loss) for different tasks (CDM, BDM, and segmentation). Ablation experiments for these tasks have been presented in Sec. 4.3 (Ablation Studies on Tasks). In this response, we focus specifically on the weighting of the BCE loss and IoU loss in the **Appendix A.5**. The results suggest that variations in performance across different weight configurations are marginal. As a result, we empirically adopted a 1:1 weight ratio. Here we provide the results in following table.
>
> **Table 1** Ablation study on the weight setting of loss functions.
>
> | ID  | Configuration                          | Fωβ | M    | Eξ | Sα | mDice | mIoU |
> | --- | -------------------------------------- | ----------------------- | ---- | ------------- | ------------ | ------ | ---- |
> | #1  | LWIoU + 0.5 * LWBCE | 0.898                   | 0.056 | 0.913         | 0.872        | 0.899  | 0.839 |
> | #2  | 0.5 * LWIoU + LWBCE | 0.893                   | 0.057 | 0.909         | 0.864        | 0.892  | 0.831 |
> | #3  | 0.5 * LWIoU + 0.5 * LWBCE | 0.899                   | 0.056 | 0.911         | 0.873        | 0.898  | 0.837 |
> | #4  | LWBCE                       | 0.867                   | 0.062 | 0.901         | 0.837        | 0.878  | 0.803 |
> | #5  | LWIoU                       | 0.824                   | 0.103 | 0.857         | 0.804        | 0.829  | 0.783 |
> | #6  | LWIoU + LWBCE    | 0.902                   | 0.056 | 0.916         | 0.875        | 0.902  | 0.841 |
> |  |

---

> > ### Author Response · Authors · 2024-11-25
> >
> > **Q3:  The paper does not explain the advantages of this one-step segmentation over the prior visual reconstruction method, and the experiment does not compare it with another method.**
> >
> > We sincerely appreciate your insightful feedback. The one-step segmentation method we propose directly segments object from lensless measurements, bypassing the intermediate reconstruction stage. This design not only reduces computational overhead but also avoids the risk of task interference, where the weak performance of one task (reconstruction or segmentation) could adversely affect the other. We have added a description of this advantage in the revised manuscript (**Appendix A.8**).
> >
> > While the focus of this paper is on the architecture and its potential benefits, we acknowledge the importance of comparing our method with traditional reconstruction-based methods. To this end, we employ FlatNet to reconstruct the underlying scene, followed by segmentation using methods such as CDMNet, BDG-Net, and ZoomNet. Our method, however, retains its original configuration. The comparative results clearly demonstrate that our method outperforms these state-of-the-art methods in segmentation accuracy. To further clarify, we have included a detailed analysis of "reconstruction + segmentation" methods in the **Appendix A.8**.
> >
> > In a nutshell, the key advantage of our method lies in its one-step design, which avoids the complexities and potential errors associated with reconstruction, thus improving overall segmentation performance and reducing the impact of reconstruction bottlenecks.
> >
> > **Q4: There is a lack of a more detailed description of the datasets. According to my understanding, are these datasets all synthetic? Are the measurements of the images synthesized using prior knowledge?**
> >
> > We sincerely appreciate your inquiry. The datasets and measurements used in our study are **shoted by PHlatCam, rather than being synthesized**. Below is a detailed description of the datasets:
> >
> > - **DISC dataset:** It includes 5.9K lensless measurements (500×620×4) captured from an LCD display, paired with 5.9K scenario images and annotation maps (256×256×3). The DISC dataset spans 869 categories (including flying, aquatic, terrestrial, amphibian, sky, vegetation, indoor, and others), with each category containing between 1 and 10 scenarios. The dataset is randomly split into training and testing sets: 5.2K images are used for training (DISC-Train), covering 744 categories (up to 10 images per category), while 0.7K images are used for testing (DISC-Test), with 473 categories (up to 3 images per category).
> > - **DIRC dataset:** It consisting of real-world natural scenes directly captured by PHlatCam. It contains 30 data pairs, each consisting of lensless measurements (500×620×4) and corresponding scenario images and annotation maps (256×256×3), sourced from 10 different scenes. The DIRC dataset is used for testing and evaluating the performance of the proposed method in real-world conditions.
> >
> > These datasets allow us to demonstrate the feasibility and generalizability of our method in performing segmentation tasks tailored for lensless imaging. To clarify this point, we provide a detailed explanation in the **Appendix A.2**.
> >
> > The comments in "Question" are identical to those in"Weaknesses." Therefore, we kindly refer to our responses in "Weaknesses" and will not provide a separate reply to the comments in "Question."

---

> > > ### Comment · Reviewer_8fG3 · 2024-11-26
> > > **response**
> > >
> > > Thanks for your explanation, it solved my concerns so I will improve my rating.

---

> > > > ### Author Response · Authors · 2024-11-26
> > > >
> > > > We sincerely thank you for your recognition of our work, which is a great encouragement to us. We will continue striving to advance this research area with dedication and innovation. Thanks again.

---

> > > ### Comment · Area_Chair_roFj · 2024-11-26
> > > **PhlatCam dataset?**
> > >
> > > Can you please elaborate how did you use phlatcam dataset with a separable system?

---

> ### Author Response · Authors · 2024-12-03
>
> Thanks for your question. In our work, we only used data provided by FlatCam, which is a separable system. On the other hand, PHlatCam is not a separable system, so its collected data cannot be directly applied to our method. However, we can adapt the data from PHlatCam by adjusting the representation of the OFD to a Wiener filter-based formulation, as :$X_\theta = \mathcal {F}^{-1}\left(\frac{ \left(\mathcal{F}(A_\theta)\right)^\top}{K_\theta+|\mathcal{F}(A_\theta)|^2} \odot \mathcal {F}(Y_\theta)\right)$, where $A_\theta$ is the PSF,  $\mathcal {F}$ and $\mathcal {F}^{-1}$ are the Fourier Transform and its inverse. $K_\theta$ is the regularization parameter.

---

### Official Review · Reviewer_Jnbq · 2024-10-29

**Soundness:** 3
**Presentation:** 3
**Contribution:** 3
**Rating:** 6
**Confidence:** 3

**Summary:**

To enhance segmentation accuracy while ensuring privacy, the authors propose a one-step method called FDTDNet for lensless object segmentation from lensless measurements without visual reconstruction. They propose an optical-aware feature demultiplexing (OFD) mechanism aimed at refining the features obtained from lensless measurements via modeling the linear equation between the semantic features bound to lensless measurements and those corresponding to visual inputs. They decouple the segmentation task into a contour distribution map (CDM) and a body distribution map (BDM) inference by contour-/bodydistribution learning branches, and propose a contour-body interaction (CBI) module for reasoning segmentation results from correlations between CDM and BDM. They conducted extensive experiments to verify their methods.

**Strengths:**

The originality is supported by modelling the linear equation between the semantic features bound to lensless measurements and those corresponding to visual inputs, and application of multiple current machine learning methods to a new domain, i.e., lensless object segmentation. The quality, clarity and significance of this work is good.

**Weaknesses:**

Equation 1 is the basis for their modeling and derivation of the relationship between the original image and the measurement in the feature space. However, Equation 1 itself is not convincing. That is, does the linearity between the original image and the measurement mean that the semantic features of the original image and the measurement are also linear? The authors should have a more rigorous derivation or proof for this.

**Questions:**

In OFD, one downsampling and two CBRs are used to transform $A_L $ or $A_R$ into its semantic space, and one PVT is used to transform the measurement Y into its semantic space. Why not do the same for AL/AR and Y? What is the author's consideration?

---

> ### Author Response · Authors · 2024-11-25
>
> We sincerely appreciate your insightful comments and recognition of our work. To ensure clarity and thoroughly address your concerns, we have provided detailed responses to each comment individually.
>
> **Weaknesses:**
>
> **Q1:does the linearity between the original image and the measurement mean that the semantic features of the original image and the measurement are also linear?**
>
> We greatly appreciate your attention to Eq. (1) in our paper. While the linearity of the semantic features between the original image and the measurements may not always hold perfectly, it serves as an effective approximation in our model. This simplification enables computational efficiency while still providing accurate feature recovery. Our hypothesis builds on the foundational work in Eq. (1) and Eq. (2) of reference [1], which extends the model from the image domain to the feature domain, achieving improved performance. Following this rationale, we have incorporated a similar assumption into our framework, as demonstrated in Eq. (1). The experimental results further validate the effectiveness of this design.
>
> [1]Dong, Jiangxin, Stefan Roth, and Bernt Schiele. "DWDN: Deep Wiener deconvolution network for non-blind image deblurring." IEEE Transactions on Pattern Analysis and Machine Intelligence, 44.12 (2021): 9960-9976.
>
> **Questions:**
>
> **Q1:Why not do the same for AL/AR and Y? What is the author's consideration?**
>
> We greatly appreciate your insightful comments and concerns regarding the design of our OFD. The distinct treatment of $Y$, $A_L$ and $A_R$ is attributed to their differing roles in the system. $Y$ encapsulates rich semantic information related to the scene, while $A_L$ and $A_R$ are primarily associated with the system's transfer function and do not inherently contain scene-specific information, limiting their semantic content. Consequently, $Y$ requires a more sophisticated feature extractor to effectively capture its semantic characteristics, whereas $A_L$ and $A_R$ can be processed with simpler convolutional operations, reducing network complexity while maintaining efficiency. This design choice strikes a balance between computational demands and the need for accurate feature extraction.

---

### Official Review · Reviewer_4SmU · 2024-11-03

**Soundness:** 2
**Presentation:** 3
**Contribution:** 3
**Rating:** 5
**Confidence:** 3

**Summary:**

This paper presents FDTDNet, a framework for object segmentation using lensless cameras, designed to enhance privacy by bypassing visual image reconstruction.

**Strengths:**

1) Quality: All figures and tables are well-designed and of high quality, except Figure 2, which will be discussed in the weaknesses section below.
2) Performance: Experiments across two different datasets validate the method’s performance. this proposed approach consistently outperforms competing methods.

**Weaknesses:**

Updated Review

Firstly, I would like to thank the authors for the clarifications and modifications provided during the discussion period. Your detailed explanations regarding the mathematical equations, as well as the associated adjustments, have addressed my initial confusion to some extent. Upon closer re-examination, it has become clear that many of the equations in the paper are existing, well-established results rather than novel contributions. While these equations may be important to your implementation, they do not appear to represent theoretical innovations. I strongly recommend that the authors explicitly highlight their contributions and clearly distinguish them from prior work to improve clarity on this point.

In addition to these observations, I have identified other concerns, including some raised by other reviewers, which I believe are more critical and warrant further discussion:

1. Overclaims on Privacy Protection:
- As highlighted by other reviewers, the privacy-preserving aspect of the proposed method seems overstated.
- While the idea of bypassing visual reconstruction aligns with privacy goals, the OFD block appears to perform some level of visual reconstruction at varying scales, which undermines the claim of mitigating sensitive privacy leakage.
- Moreover, privacy protection is presented as a core contribution, yet this aspect feels secondary or incidental to the main framework.
- Additionally, alternative imaging methods, such as single-pixel imaging or minimalist cameras, are capable of achieving similar or better privacy-preserving effects. These methods are neither discussed nor compared, which weakens the claimed contribution in this area.
- I just noticed that this point has already been addressed by the authors through revisions, so it does not require excessive concern. However, it should be noted that the contribution has been further weakened as a result, making it even more important for the authors to clearly articulate their innovations and contributions, as well as how their method differs from existing approaches.

2. Dataset Limitations and Lack of Real-World Experiments:
- The lack of real-world experiments.
- As pointed out by the AC, the datasets used in the paper are limited to controlled conditions (e.g., FlatCam/PhlatCam captures with clear foreground-background separation). This constrained setup does not sufficiently demonstrate the robustness or generalizability of the proposed method to complex, real-world scenarios, such as cluttered backgrounds, occlusions, or diverse illumination conditions.
- Without such evidence, it is difficult to assess whether the method is robust enough for broader applications.

3. Dataset Renaming and Reporting Discrepancies:
- As pointed out by the AC, the datasets were renamed. The renaming of datasets (e.g., DISC, DIRC) without proper justification raises concerns about the transparency and rigor.
- Furthermore, as also pointed out by the AC, discrepancies in the reported results for competing methods (e.g., RecSegNet) compared to their original papers call into question the reliability of the reported comparisons. These issues must be clarified to ensure confidence in the findings.

Given the above concerns, I am lowering my score due to the recognition of more significant flaws of this paper. Specifically:

1. Overclaims regarding privacy protection (I just noticed that this point has already been addressed by the authors through revisions, so it does not require excessive concern. However, it should be noted that the contribution has been further weakened as a result, making it even more important for the authors to clearly articulate their innovations and contributions, as well as how their method differs from existing approaches.)
2. Insufficient experimental validation on real-world datasets
3. Transparency and rigor issues related to dataset naming and reported results

If the authors can address these points convincingly, I am willing to adjust my rating back to a positive recommendation.


_____________________________

Initial Review

This paper has two main issues:

1) Clarity:
- The equations are overly complex. The mathematical presentation, particularly in the OFD mechanism on page 3, lines 162-215 (Equations 3-11), is overly dense and challenging to understand.
- This section mainly stacks equations without sufficient explanation, making it difficult for readers to grasp the underlying principles. It would be beneficial to include more intuitive or conceptual explanations alongside these equations.
- Additionally, labeling elements of Figure 2 to indicate which parts correspond to specific equations could greatly improve clarity. Given the length and complexity of this section, I suggest either simplifying the equations or providing clearer explanations.

2) Analysis:
- The paper could benefit from a more in-depth discussion of its limitations.
- Although some failure cases are illustrated in Figure 12 on page 16 (Appendix), it would be helpful to place these directly in the main text and discuss potential solutions more explicitly.
- Discussion about addressing these limitations directly within the main body is suggested.
- However, this is a minor suggestion. My main concern is the first point about clarity.

**Questions:**

1) Can the authors provide further insights into how the method might generalize to more complex datasets, particularly in scenarios where small objects or highly cluttered backgrounds are present?
2) How does the proposed FDTDNet handle noise in real-world lensless measurements? Could additional noise abatement strategies enhance the robustness of the segmentation?
3) Could the authors expand on the potential for adapting the method to edge devices, considering the computational demands highlighted in the complexity analysis?

---

> ### Author Response · Authors · 2024-11-25
>
> We sincerely thank you for your valuable comments and recognition, and have provided individual responses to address each concern clearly and thoroughly.
>
> **Weaknesses:**
>
> **Q1：The clarity of our manuscript.**
>
> We appreciate your valuable feedback regarding the clarity of our manuscript. To address your concerns thoroughly, we have responded to each point individually:
>
> - **About the mathematical presentation.** We appreciate your point regarding the complexity of the mathematical presentation, particularly in the OFD mechanism. While our intention was to provide a rigorous and detailed explanation, we recognize that the density of Eqs.(3)-(11) may hinder clarity. In the revised manuscript, we have simplified the mathematical formulation by merging the steps and providing more intuitive explanations for key concepts, as shown in **Appendix A.1 and Sec. 3.2**. Our goal is to make the equations more accessible while maintaining the technical precision of the proposed method.
> - **This section mainly stacks equations without sufficient explanation, making it difficult for readers to grasp the underlying principles.** We agree that the section would benefit from clearer explanations. In the revision, we include more intuitive and conceptual descriptions alongside the Eqs.(3)-(11) to better highlight the underlying principles and enhance accessibility for readers, as shown in **Appendix A.1 and Sec. 3.2.**
>
> -**Labeling elements of Figure 2 to indicate which parts correspond to specific equations could greatly improve clarity.** Thanks for your helpful suggestion about Fig.2. We have labeled the elements of the figure to clearly correspond with the relevant equations, aiming to improve the clarity of the presentation. Additionally, we have simplified the equations where possible and provided clearer explanations to enhance readability while maintaining technical accuracy.
>
> **Q2: The analysis of proposed method.**
>
> We greatly appreciate your concerns regarding the analysis of our experiments. We respond to each point as follows:
>
> - **The paper could benefit from a more in-depth discussion of its limitations.** Thanks for your feedback. While we have addressed limitations in Appendix A.5, we agree that this discussion could be expanded. We have included more detailed textual descriptions of the limitations in **Sec. 4.4** of the revised manuscript.
> - **Although some failure cases are illustrated in Figure 12 on page 16 (Appendix), it would be helpful to place these directly in the main text and discuss potential solutions more explicitly.** We agree that placing the failure cases directly in the main text, along with a more explicit discussion of potential solutions, will enhance clarity. In the revision, we have integrated the failure cases from Fig. 12 into the main text and provide a detailed analysis of these cases, including possible solutions and improvements, as illustrated in **Sec. 4.4**.
> - **Discussion about addressing these limitations directly within the main body is suggested.** We agree and have incorporated a detailed discussion of the limitations and potential solutions directly into the main text for better clarity, as illutrasted in **Sec. 4.4** of the revised manuscript.
> - **However, this is a minor suggestion. My main concern is the first point about clarity.** Thanks for your feedback. We have prioritized improving clarity in the revised manuscript to address your main concern.

---

> > ### Author Response · Authors · 2024-11-25
> >
> > **Questions:**
> >
> > **Q1:Can the authors provide further insights into how the method might generalize to more complex datasets, particularly in scenarios where small objects or highly cluttered backgrounds are present?**
> >
> > Thank you for your insightful question. Currently, our method faces challenges when applied to more complex datasets, particularly those with small objects or cluttered backgrounds, due to both the inherent limitations of lensless imaging and the increased difficulty of segmentation in such environments. To improve generalization in these cases, we can consider techniques such as **multi-scale feature extraction** and  **background suppression** , which can help capture finer details and reduce the impact of clutter. **Context-aware segmentation** methods, which adapt to local spatial variations, could also help improve performance in these complex environments. Furthermore, **domain adaptation** or **transfer learning** methods could be explored to align our model with more complex datasets, enabling better performance in such challenging conditions. Incorporating such strategies would likely improve our model’s ability to handle challenging segmentation tasks, making it more robust to variations in object size and scene complexity.
> >
> > **Q2:How does the proposed FDTDNet handle noise in real-world lensless measurements? Could additional noise abatement strategies enhance the robustness of the segmentation?**
> >
> > Thanks for your question. Our method does not explicitly address noise, as the areas targeted for segmentation generally correspond to high intensity, relatively flat regions in the scene, and noise impact is limited. However, we acknowledge that noise can still affect segmentation performance, especially in real-world lensless measurements. Traditional denoising techniques may inadvertently remove high-frequency information critical for accuracy. To mitigate this, we plan to integrate a frequency band selection mechanism into the OFD module, which would help filter out noise while preserving key high-frequency details. This strategy has the potential to enhance the robustness of segmentation in real-world noisy lensless measurements, and we will explore its effectiveness in future work.
> >
> > **Q3: Could the authors expand on the potential for adapting the method to edge devices, considering the computational demands highlighted in the complexity analysis?**
> >
> > Thanks for your insightful question. Adapting the method to edge devices is an important consideration, especially given the computational demands highlighted in our complexity analysis. While our current implementation is designed for high-performance environments, future adaptations could leverage **model compression techniques** such as  **pruning** ,  **quantization** , and **knowledge distillation** to reduce resource requirements. Additionally, **lightweight architectures** tailored for edge computing, combined with  **optimized inference pipelines** , could make deployment on resource-constrained devices feasible. We plan to explore these strategies to enable efficient execution on edge devices (such as in disease-diagnostic designs for endoscopes, microscopes, etc.) while maintaining segmentation accuracy.

---

> > > ### Comment · Reviewer_4SmU · 2024-11-27
> > >
> > > Firstly, I would like to thank the authors for the clarifications and modifications provided during the discussion period. Your detailed explanations regarding the mathematical equations, as well as the associated adjustments, have addressed my initial confusion to some extent. Upon closer re-examination, it has become clear that many of the equations in the paper are existing, well-established results rather than novel contributions. While these equations may be important to your implementation, they do not appear to represent theoretical innovations. I strongly recommend that the authors explicitly highlight their contributions and clearly distinguish them from prior work to improve clarity on this point.
> > >
> > > In addition to these observations, I have identified other concerns, including some raised by other reviewers, which I believe are more critical and warrant further discussion:
> > >
> > > 1. Overclaims on Privacy Protection:
> > > - As highlighted by other reviewers, the privacy-preserving aspect of the proposed method seems overstated.
> > > - While the idea of bypassing visual reconstruction aligns with privacy goals, the OFD block appears to perform some level of visual reconstruction at varying scales, which undermines the claim of mitigating sensitive privacy leakage.
> > > - Moreover, privacy protection is presented as a core contribution, yet this aspect feels secondary or incidental to the main framework.
> > > - Additionally, alternative imaging methods, such as single-pixel imaging or minimalist cameras, are capable of achieving similar or better privacy-preserving effects. These methods are neither discussed nor compared, which weakens the claimed contribution in this area.
> > > - I just noticed that this point has already been addressed by the authors through revisions, so it does not require excessive concern. However, it should be noted that the contribution has been further weakened as a result, making it even more important for the authors to clearly articulate their innovations and contributions, as well as how their method differs from existing approaches.
> > >
> > > 2. Dataset Limitations and Lack of Real-World Experiments:
> > > - The lack of real-world experiments.
> > > - As pointed out by the AC, the datasets used in the paper are limited to controlled conditions (e.g., FlatCam/PhlatCam captures with clear foreground-background separation). This constrained setup does not sufficiently demonstrate the robustness or generalizability of the proposed method to complex, real-world scenarios, such as cluttered backgrounds, occlusions, or diverse illumination conditions.
> > > - Without such evidence, it is difficult to assess whether the method is robust enough for broader applications.
> > >
> > > 3. Dataset Renaming and Reporting Discrepancies:
> > > - As pointed out by the AC, the datasets were renamed. The renaming of datasets (e.g., DISC, DIRC) without proper justification raises concerns about the transparency and rigor.
> > > - Furthermore, as also pointed out by the AC, discrepancies in the reported results for competing methods (e.g., RecSegNet) compared to their original papers call into question the reliability of the reported comparisons. These issues must be clarified to ensure confidence in the findings.
> > >
> > > Given the above concerns, I am lowering my score due to the recognition of more significant flaws of this paper. Specifically:
> > >
> > > 1. Overclaims regarding privacy protection (I just noticed that this point has already been addressed by the authors through revisions, so it does not require excessive concern. However, it should be noted that the contribution has been further weakened as a result, making it even more important for the authors to clearly articulate their innovations and contributions, as well as how their method differs from existing approaches.)
> > > 2. Insufficient experimental validation on real-world datasets
> > > 3. Transparency and rigor issues related to dataset naming and reported results
> > >
> > > If the authors can address these points convincingly, I am willing to adjust my rating back to a positive recommendation.

---

> ### Author Response · Authors · 2024-12-03
>
> Thank you very much for your thoughtful feedback and for recognizing the clarifications and modifications we made. We appreciate your insightful comments regarding the mathematical equations and understand your concern about distinguishing our contributions. We provide detailed responses to each of your points.
>
> **About the Theoretical Innovations.** Thank you for your comment. The point we would like to emphasize is that although the equations in our paper are based on established techniques (Tikhonov Least Squares), our key innovation lies in extending the Tikhonov Least Squares method—traditionally used for image reconstruction—into the semantic feature domain. By validating the assumptions in **Eq. (1)**, we shift the application of this technique from reconstruction to direct end-to-end inference. This change overcomes the performance bottleneck often associated with the traditional "reconstruction + inference" paradigm and provides a more efficient framework for low-quality scene inference tasks. We believe this represents a significant advancement in the field, and we will emphasize this distinction more clearly in the revised manuscript.
>
> **About Privacy Protection.** As mentioned in my response to AC, in our work, the lensless measurements are directly input into the network, which only performs semantic inversion rather than visual reconstruction. This means that the information transmitted through our network is not in the form of visual data, thereby mitigating the risk of sensitive information leakage during the network's operation. we also have ensured to highlight these innovations more explicitly in the revised version of the manuscript to make the contribution clearer for readers.
>
> Regarding privacy protection, it is not the primary focus of our work but rather an additional benefit of our approach. Our main objective is to emphasize the potential of using lensless devices for high-level inference tasks, which holds significant value for expanding the applications of lensless imaging technology.
>
> As for single-pixel imaging or minimalist cameras, to the best of our knowledge, these fields are not directly related to lensless imaging technology. The privacy protection effectiveness associated with these technologies has not yet been discussed in high-level peer-reviewed journals. To maintain the rigor of our work, we have refrained from addressing this aspect. However, we acknowledge the importance of exploring the performance of these imaging mechanisms, including downstream task performance and privacy protection, in our future research.
>
> **About Real-world Experiments.** As mentioned in our response to AC's third comment, for the real-world experiments conducted under varying conditions, we utilized scene data from FlatCam, such as the DIRC dataset. This dataset includes lensless imaging measurements captured under different illumination scenarios, which partially capture the complexity of real-world environments. However, we acknowledge that the availability of publicly accessible datasets for lensless imaging remains limited, presenting challenges in validating our method across more complex scenarios. We fully agree that incorporating a broader range of datasets would provide a more robust evaluation of our approach. As lensless imaging is an emerging field, we anticipate that the availability of diverse datasets will naturally increase with its development. In the meantime, we have made every effort to include the most comprehensive data and experiments available at present, and we kindly ask for your understanding in this regard.
>
> In terms of experimental setup, while it is true that FlatCam captures are typically conducted in controlled environments, our experiments were not limited to isolated objects. For example, the DISC-Test dataset includes scenarios with multiple objects. In the revised manuscript (**Appendix Fig. 16**), we present multi-object segmentation results that highlight the strong performance of our method even in the presence of multiple objects. These results validate the potential of our approach for handling dense scenes and complex segmentation tasks. As such, we believe our findings demonstrate the significant applicability of our method to real-world environments, dense scenes, and multi-object scenarios.
>
> We hope that this response addresses your concerns and provides clarity on the scope and contributions of our work. Thank you once again for your valuable input.

---

> > ### Author Response · Authors · 2024-12-03
> >
> > **Dataset Renaming and Reporting Discrepancies.** As stated in our response to AC's third comment, regarding the naming conventions for the datasets, we primarily utilized the display capture dataset and the direct capture dataset. Lensless imaging measurements and corresponding ground-truth scenes were selected from the publicly available FlatCam dataset, which includes 5.9k samples for the display capture dataset and 30 samples for the direct capture dataset. The segmentation labels for these datasets were derived from [1]. To ensure consistency, we adopted the dataset naming conventions introduced in [1]—namely, Direct Capture (DIRC) and Display Capture (DISC) datasets—and have appropriately cited this reference.
> >
> > In terms of reporting discrepancies, we would like to clarify the reasons behind the observed differences between the results for LOINet and RecSegNet, as reported in our paper (Fig. 6) and the RecSegNet paper (e.g., Fig. 10):
> >
> > 1. *Methodological Differences*: In our manuscript, we introduced additional comparison methods that were not included in the RecSegNet paper, such as CDMNet, OCENet, LL_T, Raw3dNet, and EyeCoD. To ensure a fair and unbiased comparison across all methods (including those overlapping with RecSegNet and those not considered in the RecSegNet paper), we re-trained all models under the same experimental conditions. The objective was to provide a comprehensive comparison of a wider set of methods, rather than directly replicating the results from the RecSegNet paper. The inclusion of these additional methods and the emphasis on maintaining consistent evaluation conditions naturally led to some differences in the results.
> > 2. *Impact of Random Initialization*: As the models were re-trained from scratch, the random initialization of weights and biases could influence the convergence behavior of the networks, which in turn affects the final results. This randomness is an inherent aspect of the training process and can contribute to variations in model performance.
> > 3. *Influence of Multi-threaded Parallel Computation*: Additionally, the use of multi-threaded parallel computation in our code introduced another potential source of variability. For instance, data partitioning, loading, and optimizations in the underlying algorithms during multi-threaded execution could result in minor differences in the outcomes.
> >
> > We would like to emphasize that the results presented in our paper are intended to reflect the performance under the specific experimental setup we employed, ensuring the reliability of the findings within the context of our study. We hope this explanation clarifies the reasons behind the observed discrepancies and provides a clearer understanding of our experimental methodology.
> >
> > Reference:
> >
> > [1] Xiangjun Yin, Huanjing Yue, Huihui Yue, Mengxi Zhang, Kun Li, and Jingyu Yang. A multi-task deep learning framework integrating segmentation and reconstruction for lensless imaging. IEEE Transactions on Emerging Topics in Computational Intelligence, 2024.

---

> ### Comment · Reviewer_4SmU · 2024-12-03
>
> Thank you for your detailed response. However, I find that my concerns regarding the innovations and contributions of your work remain unresolved. My concerns can be summarized into three main points:
>
> 1. Privacy Protection:
> From a high-level perspective, I do not see any unique or superior advantages in privacy protection offered by your approach compared to other existing methods. Even without lensless imaging, other imaging techniques, such as single-pixel imaging and minimalist cameras, can achieve similar levels of privacy protection. As a result, your work does not appear to make any distinct or original contributions in this regard.
>
> 2. Direct High-Level CV Tasks on Measurements:
> If your main contribution is the ability to perform high-level CV tasks, such as segmentation, directly on measurements, this also does not seem novel. For example, RecSegNet [1] is designed to perform both reconstruction and segmentation simultaneously rather than sequentially, so its method can also work without reconstructing the original visual data. In your response to the AC, you stated that “the OFD operates at the feature level, rather than on visual images.” However, after reading the RecSegNet paper, I found that its segmentation also operates at the feature level rather than on visual images. Additionally, other works in computational imaging have already explored performing CV tasks directly on measurements, such as [2].
>
> Therefore, whether from the perspective of privacy protection or from performing CV tasks directly on measurements, your work does not stand out as innovative or unique.
>
> 3. Scope and Contribution Level:
> Your work seems more like a targeted solution within the specific application domain of lensless imaging, addressing how to perform segmentation directly on measurements in that context. In my opinion, this type of work does not reach the level of innovation and contribution typically expected for a conference like ICLR. Furthermore, after reviewing the references in your paper and confirming related work, I noticed that most similar research is published in optics or computational imaging journals rather than at machine learning conferences like ICLR. This suggests that your work might be somewhat out of scope for ICLR.
>
> To justify its importance and relevance to ICLR, you need to make a much stronger case for the significance of your work and its connection to the broader machine learning community.
>
> [1] Yin, Xiangjun, et al. "A Multi-Task Deep Learning Framework Integrating Segmentation and Reconstruction for Lensless Imaging." IEEE Transactions on Emerging Topics in Computational Intelligence (2024).
>
> [2] Zhang, Zhihong, et al. "From compressive sampling to compressive tasking: retrieving semantics in compressed domain with low bandwidth." PhotoniX 3.1 (2022): 19.

---

> > ### Author Response · Authors · 2024-12-03
> >
> > Thank you sincerely for your response. A closer examination of RecSegNet reveals that it incorporates an Optical-aware Encoder (OE) to perform an initial reconstruction, which is cascaded before the encoder and essentially follows a conventional visual reconstruction paradigm. In contrast, our OFD module is intrinsically integrated within the encoder, representing a genuinely feature-level inversion design. We kindly request the reviewer to carefully consider this distinction.

---

> > > ### Author Response · Authors · 2024-12-03
> > >
> > > Regarding the relevance of our work to the scope of ICLR, we kindly request the reviewers to evaluate our submission within the context of the designated track, rather than dismissing it prematurely.

---

> > ### Author Response · Authors · 2024-12-03
> >
> > We kindly request that the reviewers give careful consideration to our work. Our approach is not merely a straightforward application of deep neural networks; rather, it represents an expansion of lensless imaging technology in both its application performance and method design. Lensless imaging, as a compact solution, has a broad range of potential applications, including medical endoscopy and surveillance in narrow spaces, where traditional lenses fall short. Consequently, exploring segmentation methods based on lensless imaging has become both a critical and urgent task. Our proposal aims to provide a more efficient and high-precision technical pathway for the successful deployment of downstream tasks within lensless imaging systems.
> >
> > In terms of applications, our method integrates seamlessly with lensless imaging systems, enabling high-precision image segmentation. When compared with the method proposed in [2], it is important to note that [2] is primarily focused on detection tasks, and its performance in more challenging segmentation tasks remains unverified. Moreover, [2] requires deep reconstruction, which still relies on the "reconstruction + inference" framework, limiting its flexibility and preventing the achievement of privacy protection goals. In contrast, RecSegNet [1] demands the simultaneous execution of both reconstruction and segmentation tasks, meaning that segmentation performance is inherently dependent on reconstruction, thus failing to fulfill privacy protection objectives. Furthermore, as previously mentioned, RecSegNet uses an initial reconstruction module (OE) that outputs a three-channel image hierarchy instead of multi-channel feature expressions, creating a bottleneck. This distinction highlights significant differences in the task execution process between RecSegNet and our method. Notably, our approach does not involve reconstruction of visual information at any stage from input to output, a capability that remains unachieved in most related works, including RecSegNet. We hope the AC to carefully consider these aspects during the review.
> >
> > Regarding method design, in addition to the encoder design based on OFD, we propose a task-decoupling strategy that decomposes tasks into two simpler sub-tasks tailored to the characteristics of lensless imaging, thereby enhancing performance. We hope that the AC and reviewers will recognize this contribution, rather than evaluating our work solely from the perspective of network modularity.
> >
> > Reference：
> >
> > [1] Yin, Xiangjun, et al. "A Multi-Task Deep Learning Framework Integrating Segmentation and Reconstruction for Lensless Imaging." IEEE Transactions on Emerging Topics in Computational Intelligence (2024).
> >
> > [2] Zhang, Zhihong, et al. "From compressive sampling to compressive tasking: retrieving semantics in compressed domain with low bandwidth." PhotoniX 3.1 (2022): 19.

---

### Official Review · Reviewer_Ztf8 · 2024-11-05

**Soundness:** 3
**Presentation:** 3
**Contribution:** 3
**Rating:** 6
**Confidence:** 4

**Summary:**

The paper proposes a model for segmentation that operates on measurements from a lensless camera. Instead of prior approaches that first attempt to reconstruct an RGB image and then carry out segmentation, the paper's approach directly operates on the lensless measurements. The architecture is endowed with knowledge of the optical measurement process through "optical feature demultiplexing", along with other innovations. Experimental results confirm the benefits of this approach.

**Strengths:**

- The paper is generally well motivated (except for the question of privacy below) and written. It makes sense that a single unified approach would work better than segmenting reconstructed images.
- The OFD approach is novel and interesting. It has the potential to be useful beyond the segmentation task as a general way of processing lensless measurements for vision tasks.
- The experiments and ablations are extensive and largely convincing.

**Weaknesses:**

- The paper adds an unnecessary "privacy preserving" claim (in its title!) that is really only discussed in the (first paragraph of the) introduction, and mostly by citing other papers. Privacy preserving is a strong claim and should not be made without more care. If anything, a paper that shows improved performance at segmentation implies that lensless measurements  carry a fair amount of information about the underlying scene, and could leak private details. A video of segmentation masks could, for example, be enough to identify people by gaits. At that point, we get to deciding what privacy preserving means and what kind of privacy is being preserved.

  But this entire question is un-necessary to the central contribution of the paper --- a better segmentation approach for lensless cameras. The paper would be stronger, and in my opinion more sound, if it dropped the superfluous privacy claim from its title.

- The ODM + CDM approach could be explained a bit better, and especially discussed more with related work. Has this division into subtasks been tried before? How does this relate to CDMNet?

- Minor point, but the paper should make the experimental results section a bit more self contained and describe the content of the two benchmark datasets.

**Questions:**

Please address the points brought up in the weakness section above.

---

> ### Author Response · Authors · 2024-11-25
>
> We greatly value your profound insights and acknowledgment, and have provided meticulous, point-by-point responses to address each concern with precision and clarity.
>
> **Weaknesses:**
>
> **Q1:The paper adds an unnecessary "privacy preserving".**
>
> Thank you for highlighting concerns regarding the "privacy-preserving" claim in the title. Based on your valuable feedback, we have removed this term from the title and revised the manuscript to better align with the central contribution of our work—a segmentation approach for lensless cameras.
>
> Our initial mention of "privacy-preserving" was motivated by the inherent properties of lensless imaging, where lensless measurements lack directly interpretable high-resolution details, which can reduce the risk of immediate information leakage. This aspect is particularly relevant for applications where safeguarding sensitive details is essential. However, as you rightly pointed out, positioning the paper around privacy-preserving could detract from its primary focus.
>
> We appreciate your thoughtful suggestion and have revised the discussion in the introduction to clarify this point and avoid any perception of overclaiming. Thank you again for your insightful feedback, which has helped us strengthen the clarity and focus of the manuscript.
>
> **Q2:The ODM + CDM approach could be explained a bit better, and especially discussed more with related work. Has this division into subtasks been tried before? How does this relate to CDMNet?.**
>
> Thanks for your insightful question. Our method of decoupling the segmentation task into CDM and BDM addresses the inherent imbalance in pixel distributions during segmentation. While task decomposition, such as edge-based guidance, has been explored in works like CDMNet, our division into CDM and BDM offers a novel method tailored for lensless imaging, where spatial boundaries are often ambiguous or absent. Our method extends this concept by introducing the contour-body interaction (CBI) module, which models the correlations between CDM and BDM, enabling mutual learning and enhancing segmentation accuracy. In contrast, CDMNet focuses primarily on edge-based cues and does not utilize a dual-branch interaction for more comprehensive segmentation. We appreciate your suggestion to further discuss and ensure that these connections are more clearly articulated in **Sec. 4.2** of revised manuscript.
>
> **Q3:Minor point, but the paper should make the experimental results section a bit more self contained and describe the content of the two benchmark datasets.**
>
> Thanks for your valuable suggestion. We agree that providing more context on the benchmark datasets and ensuring the experimental results section is more self-contained would enhance the clarity of the paper. In the revised manuscript, we have expanded on the characteristics and content of the DIRC and DISC datasets, detailing their specific relevance to our method and how their role in facilitating the evaluation of lensless object segmentation. This additional information will help readers better understand the experimental setup and ensure that the results are more comprehensively presented. The additional details about the two benchmark datasets can be found in  **Appendix A.2 (Page 14)** .

---

> > ### Comment · Reviewer_Ztf8 · 2024-11-26
> >
> > Thanks for the updates. On Q2 and 3, my concerns are largely resolved.
> >
> > I also appreciate that the authors have toned down the privacy claims in the abstract/intro and removed it from the title. Again, I have always viewed lensless imaging's advantage being more compact cameras (no need for a lens) rather than benefits for privacy. And I am not aware of any rigorous work showing that private/sensitive information can not be extracted from lensless measurements --- especially given that _this_ paper is about successfully extracting pretty good segmentation maps from those images. There I feel that, if the paper is accepted, mention of privacy can be toned down further.
> >
> > I'm keeping my score because I don't think the paper is at a score of an "8" --- but I'd say it's closer to 7 than 6.

---

> ### Author Response · Authors · 2024-12-03
>
> Thanks for your concerns regarding privacy protection. As mentioned in my response to AC, in our work, the lensless measurements are directly input into the network, which only performs semantic inversion rather than visual reconstruction. This means that the information transmitted through our network is not in the form of visual data, thereby mitigating the risk of sensitive information leakage during the network's operation. Thank you once again for your recognition of our work.

---

### Comment · Area_Chair_roFj · 2024-11-26
**comments from AC**

Dear authors,

I read your paper and would like to bring up some concerns for discussion. I will encourage the reviewers to add/correct me if I missed anything.

1. Privacy claims are brought up by other reviewers, and I think we will further discuss it there if needed. I would like to hear some clarification on what do you mean by "the OFD mechanism facilitates back-end tasks without visual reconstruction, mitigating sensitive privacy leakage." in L202-203?
It seems to me that your OFD block will provide a visual reconstruction at different scales. If so, please add some sample reconstructions.

2. I have another question about the so-called OFD-based extractor module. How is this new or a significant contribution that has a subsection and an appendix? It is a well-known Tikhonov Least Squares solution for a separable system, an identical version was proposed in the original FlatCam paper and subsequently used in other follow up papers.

3. One serious concern related to the lack of real experiments and the specific dataset used in this paper. I consider the lack of real experiments under different conditions a serious limitation of this work. The dataset and experiments in the paper seem limited to FlatCam or PhlatCam captures, (please clarify what dataset you used and also why are you giving these datasets new names as DISC and DIRC if you are using FlatCam or PhlatCam datasets?) Both FlatCam and PhlatCam captures images in controlled environments (illumination, isolated objects). The dataset and results are essentially foreground and background separation of single object. Is that sufficient for segmentation? How well would the proposed method work in real-world environments or dense scenes/multiple objects is unclear.

4. The results for LOINet and RecSegNet reported in this paper (Fig 6) do not match the results in (e.g., Fig 10 in ) RecSegNet paper. Why is this discrepancy?

5. When I look at this paper and RecSegNet, I feel they essentially follow same motivation and experiments with different modules for reconstruction and segmentation. Is there any fundamental innovation in the proposed work over RecSegNet or LOINet?

I am writing this note to give you an opportunity to respond to some or all of these comments before tomorrows deadline for major changes in the pdf where necessary.

You do not need to add new experiments in the pdf. We can discuss these points in the message format as well.

---

> ### Author Response · Authors · 2024-11-28
>
> Thanks for your thoughtful feedback and positive discussion on our manuscript. We have carefully considered your questions and made revisions to the manuscript accordingly. Below, we provide detailed responses to each of your points.
>
> **Q1: I would like to hear some clarification on what do you mean by "the OFD mechanism facilitates back-end tasks without visual reconstruction, mitigating sensitive privacy leakage." in L202-203?**
>
> We sincerely appreciate your insightful question regarding the statement in L202-203: "the OFD mechanism facilitates back-end tasks without visual reconstruction, mitigating sensitive privacy leakage." This provides an opportunity to further clarify our work.﻿
>
> Traditional methods to downstream tasks typically start by reconstructing visual images from lensless measurements before extracting task-relevant features. While effective, this practice poses a significant risk in privacy-sensitive scenarios, such as medical imaging or surveillance, as the reconstructed images may inadvertently reveal sensitive information—even if such details are irrelevant to the downstream task.
>
> To address this issue, our proposed OFD eliminates the need for visual image reconstruction altogether. Specifically, the OFD is designed to abtain high-level semantic features ($X_\theta$) associated with the underlying scene  from high-level semantic features ($Y_\theta$) associated with lensless measurements by feature-level inversion. These features are task-relevant abstractions rather than direct visual data. That is the OFD operates at the feature level, rather than on visual images. Therefore, by operating solely on these abstract features, the OFD avoids reconstructing or extracting any visual details of the underlying scene, thereby significantly mitigating the risk of sensitive information leakage.
>
> To further clarify, we have provided examples about the outputs of the OFD in  **Appendix Fig. 9** . These results clearly demonstrate that the outputs of the OFD are composed entirely of abstract semantic features, such as object contours, which are effective for downstream task performance while remaining devoid of sensitive visual details. This reinforces the privacy-preserving nature of our method, as it circumvents the reconstruction of original visual data.
>
> We hope this clarification, along with the additional context provided in  **Appendix Fig. 9** , enhances your understanding of how the OFD mechanism operates and safeguards privacy.
>
> **Q2: I have another question about the so-called OFD-based extractor module. How is this new or a significant contribution that has a subsection and an appendix? It is a well-known Tikhonov Least Squares solution for a separable system, an identical version was proposed in the original FlatCam paper and subsequently used in other follow up papers.**
>
> Thanks for your insightful question. While the Tikhonov Least Squares is a well-established technique, and was effectively applied in the original FlatCam paper, our work introduces meaningful advancements. In traditional methods, Tikhonov Least Squares is typically used for image reconstruction. However, our method extends this technique to the semantic feature level, shifting its application from the image level to the feature level. This innovation not only eliminates reconstruction errors that typically hinder task performance in traditional methods but also significantly enhances the efficacy of downstream tasks. Consequently, the Tikhonov Least Squares solution discussed in our paper—especially in the dedicated subsection and appendix—is applied specifically at the feature level.
>
> Furthermore, in the OFD module, unlike the traditional Tikhonov Least Squares solution that integrates $A_L$ and $A_R$ in a fixed manner, our method incorporates a feature extraction module to derive learnable $A_{L,\theta}$ and $A_{R,\theta}$. This enables them to adapt to the specific needs of downstream tasks, making the application of Tikhonov Least Squares more flexible and efficient within our framework.
>
> Finally, the multi-level architecture of the OFD provides a richer semantic features for applying Tikhonov Least Squares at the feature level. By applying Tikhonov Least Squares across multiple scales, we effectively integrate complementary information, enhancing robustness and reducing errors, which further improves performance.
>
> To clearly explain the differences and innovations of our method compared to traditional Tikhonov Least Squares, we used a subsection and an appendix, where we provide a detailed explanation of our method’s unique contributions. We believe these improvements represent a meaningful advancement over existing methods.

---

> ### Author Response · Authors · 2024-11-28
>
> **Q3:One serious concern related to the lack of real experiments and the specific dataset used in this paper.**
>
> Thanks for your thoughtful and constructive feedback. We appreciate the opportunity to address your concerns regarding the datasets and experiments in our work.
>
> For real experiments conducted under varying conditions, we utilized real-world scene data provided by FlatCam, such as the DIRC dataset. This dataset includes lensless imaging measurements captured under different illumination conditions, which partially reflect the complexity of real-world scenarios. However, we acknowledge that the availability of publicly accessible datasets for lensless imaging remains limited, making it challenging to verify our method under more complex scenarios. We fully agree that incorporating more diverse datasets would better validate the robustness of our approach. As lensless imaging is still an emerging field, we believe the availability of datasets will naturally grow as the field develops. We have made every effort to include the most comprehensive data types and experiments currently available in our manuscript and kindly request your understanding on this matter.
>
> Regarding the source and naming of datasets, they primarily encompass the display capture dataset and the direct capture dataset. Lensless imaging measurements and paired ground-truth scenes were selected from the publicly available FlatCam dataset (5.9k samples for the display capture dataset and 30 samples for the direct capture dataset), with corresponding segmentation labels derived from [1]. To ensure consistency, we adopted the dataset naming conventions introduced in [1] (i.e, Direct Capture (DIRC) dataset and the Display Capture (DISC) dataset) and cited this reference accordingly.
>
> For the experimental setup, while it is true that FlatCam captures are often conducted in controlled environments, our experiments were not limited to isolated objects. For instance, the DISC-Test dataset includes scenarios with multiple objects. In the revised manuscript ( **Appendix Fig. 16** ), we present multi-object segmentation results that demonstrate the strong performance of our method even in the presence of multiple objects. These results validate the potential of our approach for dense scenes and more complex segmentation tasks. Consequently, we believe our findings highlight the significant applicability of the proposed method to real-world environments, dense scenes, and multi-object scenarios.
>
> We hope this response addresses your concerns and clarifies the scope and contributions of our work. Thank you again for your valuable input.
>
> Reference
>
> [1] Xiangjun Yin, Huanjing Yue, Huihui Yue, Mengxi Zhang, Kun Li, and Jingyu Yang. A multi-task deep learning framework integrating segmentation and reconstruction for lensless imaging. IEEE Transactions on Emerging Topics in Computational Intelligence, 2024.
>
> **Q4:The results for LOINet and RecSegNet reported in this paper (Fig 6) do not match the results in (e.g., Fig 10 in ) RecSegNet paper. Why is this discrepancy?**
>
> Thanks for raising this important question. I would like to clarify the reasons behind the observed discrepancy between the results for LOINet and RecSegNet reported in this paper (Fig. 6) and those in the RecSegNet paper (e.g., Fig. 10):
>
> 1) In our manuscript, we have introduced additional comparison methods that were not included in the RecSegNet paper, such as CDMNet, OCENet, LL_T, Raw3dNet, and EyeCoD. To ensure a fair and unbiased comparison across all methods (including those overlapping with the RecSegNet paper and those not included in it), we re-trained all models under a same experimental conditions. Our goal was to provide a comprehensive comparison of a broader set of methods, rather than directly replicate the results reported in the RecSegNet paper. The inclusion of these additional methods and the emphasis on maintaining consistent evaluation conditions naturally led to some differences in the results.
>
> 2) As the models were re-trained from scratch, the random initialization of weights and biases can influence the convergence behavior of the networks, which in turn affects the final results. This randomness is inherent to the training process and can contribute to variations in performance.
>
> 3) Furthermore, the multi-threaded parallel computation used in codes introduces another source of randomness. For instance, in multi-threaded mode, data partitioning and loading, as well as optimizations in the underlying algorithms, may introduce minor differences in the results.
>
> We would like to emphasize that the results we report are intended to reflect the performance under our experimental setup, ensuring the reliability of the results in the context of our study. We hope this explanation clarifies the reasons behind the observed discrepancies and provides a better understanding of our experimental method.

---

> ### Author Response · Authors · 2024-11-28
>
> **Q5:Is there any fundamental innovation in the proposed work over RecSegNet or LOINet?**
>
> Thanks for your insightful question. To clarify the fundamental innovation of our proposed work over existing methods such as RecSegNet and LOINet, the key distinction lies in the elimination of the visual reconstruction step, providing the unified framework, and design of task decoupling and interaction , which is typically employed in these methods.
>
> 1) **No Visual Reconstruction**: Both RecSegNet and LOINet rely on a simple visual inversion setup, which inevitably introduces reconstruction errors that limit the performance of downstream tasks. These errors not only degrade task performance but also create a bottleneck in the optimization process, as the initial reconstruction and downstream tasks are somewhat isolated due to the nature of the reconstruction framework.
>
>    In contrast, our method avoids the need for visual reconstruction altogether. By directly using lensless imaging measurements in the computational framework for downstream tasks, we bypass the error accumulation introduced by reconstruction and eliminate the bottleneck caused by separating the reconstruction and task optimization processes. This method streamlines the process and facilitates more effective optimization, which results in better overall performance.
> 2) **Unified Framework**: We extend the framework paradigm for downstream tasks from traditional imaging domain to computational imaging domain, enabling a more cohesive method. Specifically, our design offers insights into performing downstream tasks in challenging imaging environments, such as low light, blurriness, noise, and extreme weather conditions. For these, we simply need to adapt the configuration of the OFD for effective application in these scenarios.
> 3) **Task Decoupling and Information Interaction**: We also design a task decoupling mechanism that improves information interaction between different components. This design enhances the segmentation performance by ensuring that task-specific information is more effectively shared and optimized across the network, which further strengthens the robustness of our method in real-world conditions.
>
> Unlike the methods in RecSegNet and LOINet primarily focus on the task of segmentation with initial reconstruction, our method offers a fundamentally different framework by eliminating the need for reconstruction and addressing the challenges inherent in lensless imaging for downstream tasks. This shift in paradigm provides a significant contribution, rather than simply adding new modules to existing systems.

---

> ### Comment · Reviewer_4SmU · 2024-12-03
>
> After carefully reviewing and reflecting on your discussion and verifying the related literature, I would like to raise a concern regarding the novelty of the authors' contribution. Specifically, I found that RecSegNet [1] is designed to perform both reconstruction and segmentation simultaneously rather than sequentially, meaning its method can also work without reconstructing the original visual data.
>
> The authors stated that their innovation and contribution lie in their OFD "operating at the feature level, rather than on visual images.” However, after reading the RecSegNet paper [1], I found that its segmentation also operates at the feature level rather than on the original visual images.
>
> Given these findings, I believe the authors need to more clearly articulate how their work demonstrates sufficient innovation and contribution compared to prior approaches.
>
> [1] Yin, Xiangjun, et al. "A Multi-Task Deep Learning Framework Integrating Segmentation and Reconstruction for Lensless Imaging." IEEE Transactions on Emerging Topics in Computational Intelligence (2024).

---

> > ### Author Response · Authors · 2024-12-03
> >
> > Thank you sincerely for your response. A closer examination of RecSegNet reveals that it incorporates an Optical-aware Encoder (OE) to perform an initial reconstruction, which is cascaded before the encoder and essentially follows a conventional visual reconstruction paradigm. In contrast, our OFD module is intrinsically integrated within the encoder, representing a genuinely feature-level inversion design. We kindly request the reviewer to carefully consider this distinction.

---

> > > ### Author Response · Authors · 2024-12-03
> > >
> > > More importantly, our work demonstrates substantial advancements in both performance and efficiency, contributing valuable insights to the advancement of lensless imaging technology. We sincerely hope the AC will undertake a rigorous evaluation of our submission. Thank you very much for your consideration. We sincerely hope that the AC and reviewers recognize our dedicated efforts in this emerging field and provide a rigorous and thoughtful evaluation of our work. Your careful consideration would be deeply appreciated.

---

> > > > ### Author Response · Authors · 2024-12-03
> > > >
> > > > Additionally, we would like to draw the reviewers' attention to the fact that our design does not suffer from a bottleneck layer, unlike other approaches (such as RecSegNet) that invariably rely on a simple sequence of initial visual reconstruction followed by downstream tasks. In these methods, the two modules are linked through a narrow bottleneck layer formed by a limited number of channels, which restricts the flow of information and, in turn, limits both representational capacity and overall performance.
> > > >
> > > > In contrast, our OFD-based feature extraction approach effectively eliminates this bottleneck, allowing for more efficient information extraction and, consequently, better performance in downstream tasks. Moreover, by adjusting the mathematical formulation of the OFD (particularly the system matrix), our method is adaptable to downstream tasks in other low-quality scenarios. We hope the reviewers and AC will give careful consideration to this insight.

---

> > ### Author Response · Authors · 2024-12-03
> >
> > We kindly request that the reviewers give careful consideration to our work. Our approach is not merely a straightforward application of deep neural networks; rather, it represents an expansion of lensless imaging technology in both its application performance and method design. Lensless imaging, as a compact solution, has a broad range of potential applications, including medical endoscopy and surveillance in narrow spaces, where traditional lenses fall short. Consequently, exploring segmentation methods based on lensless imaging has become both a critical and urgent task. Our proposal aims to provide a more efficient and high-precision technical pathway for the successful deployment of downstream tasks within lensless imaging systems.
> >
> > In terms of applications, our method integrates seamlessly with lensless imaging systems, enabling high-precision image segmentation. When compared with the method proposed in [2], it is important to note that [2] is primarily focused on detection tasks, and its performance in more challenging segmentation tasks remains unverified. Moreover, [2] requires deep reconstruction, which still relies on the "reconstruction + inference" framework, limiting its flexibility and preventing the achievement of privacy protection goals. In contrast, RecSegNet [1] demands the simultaneous execution of both reconstruction and segmentation tasks, meaning that segmentation performance is inherently dependent on reconstruction, thus failing to fulfill privacy protection objectives. Furthermore, as previously mentioned, RecSegNet uses an initial reconstruction module (OE) that outputs a three-channel image hierarchy instead of multi-channel feature expressions, creating a bottleneck. This distinction highlights significant differences in the task execution process between RecSegNet and our method. Notably, our approach does not involve reconstruction of visual information at any stage from input to output, a capability that remains unachieved in most related works, including RecSegNet. We hope the AC to carefully consider these aspects during the review.
> >
> > Regarding method design, in addition to the encoder design based on OFD, we propose a task-decoupling strategy that decomposes tasks into two simpler sub-tasks tailored to the characteristics of lensless imaging, thereby enhancing performance. We hope that the AC and reviewers will recognize this contribution, rather than evaluating our work solely from the perspective of network modularity.
> >
> > Reference：
> >
> > [1] Yin, Xiangjun, et al. "A Multi-Task Deep Learning Framework Integrating Segmentation and Reconstruction for Lensless Imaging." IEEE Transactions on Emerging Topics in Computational Intelligence (2024).
> >
> > [2] Zhang, Zhihong, et al. "From compressive sampling to compressive tasking: retrieving semantics in compressed domain with low bandwidth." PhotoniX 3.1 (2022): 19.

---

### Note · Authors · 2025-03-14

I have read and agree with the venue's withdrawal policy on behalf of myself and my co-authors.

---

### Meta-Review · Area_Chair_roFj · 2024-12-20

**Metareview:**

Summary. The paper proposes a model for segmentation that operates on measurements from a lensless camera. Instead of prior approaches that first attempt to reconstruct an RGB image and then carry out segmentation, the paper's approach directly operates on the lensless measurements.

Strengths. The paper is generally well motivated and written. The idea of segmentation on lensless measurements without reconstruction has some advantages. The experiments demonstrate good performance on two datasets.

Weaknesses. The paper adds unsubstantiated "privacy preserving" claims. Technical innovations of the paper are limited and incremental. The OFD-based extractor module uses a Tikhonov least squares solution fo a separable system, which authors claim as a novel contribution. The datasets and experiments in the paper are limited to two datasets (mainly used for image reconstruction, but repurposed by authors for segmentation), which do not represent realistic/complex scenes needed for segmentation. Results of prior work presented in the paper are inconsistent with the published work. The proposed segmentation method bears strong resemblance with LOINet and RecSegNet.

What is missing.
Authors proposed to tone-down privacy-preserving claims during the rebuttal, but insist on that claim without convincing answers to questions and concerns raised by the reviewers.
Technical innovation of the proposed method is unclear. The OFD module combines Tikhonov solution in FlatCam with learnable matrices in FlatNet. Segmentation modules follow the framework proposed in LOINet and RecSegNet.
Experiments for real scenes mainly use data captured by FlatCam and PhlatCam, where scenes largely consist of a single object with black background. Segmentation on more realistic scenes is essential to demonstrate the utility of the proposed method.
Authors offered contradictory explanations on using FlatCam and PhlatCam datasets. FlatCam assumes a separable model, while PhlatCam assumes convolutional model. This part should be clarified in any revision.

Justification.
Privacy-preserving claims are not justified. The paper lacks technical novelty and experiments do not appear significant. The improvements and advantages over similar lensless-segmentation methods is unclear.

**Additional Comments On Reviewer Discussion:**

The paper was discussed extensively among the authors, reviewers, and AC.
The reviewers raised questions about the claims of novelty and privacy-preserving, lack of real-world experiments, lack of explanation on how the proposed method differs or improves existing methods.

Authors provided detailed responses that clarified some aspects, but could not convince reviewers on three aspects: privacy-preserving claims, technical novelty of the proposed method over existing methods, and significance of results in the absence of real-world experiments. AC agrees with these concerns.

Reviewers had a good discussion after the rebuttal period. Those who participated in the discussion lean toward reject.

---

### Decision · Program_Chairs · 2025-01-22

Reject